

# Interannual variations of early winter Antarctic polar stratospheric cloud formation and nitric acid observed by CALIOP and MLS

A. Lambert[1], M. L. Santee[1], and N. J. Livesey[1]

[1]Jet Propulsion Laboratory, California Institute of Technology, Pasadena, California, USA

*Correspondence to:* A. Lambert, (Alyn.Lambert@jpl.nasa.gov)

**Abstract.**

We use satellite-borne measurements collected over the last decade (2006-2015) from the Aura Microwave Limb Sounder (MLS) and the Cloud-Aerosol Lidar with Orthogonal Polarization (CALIOP) to investigate the nitric acid distribution and the properties of polar stratospheric clouds (PSCs) in the early winter Antarctic vortex. Frequently, at the very start of the winter, we

find that synoptic-scale depletion of $HNO_3$ can be detected in the inner vortex before the first lidar detection of geophysically associated PSCs. The generation of "sub-visible" PSCs can be explained as arising from the development of a solid particle population with low number densities and large particle sizes. Assumed to be composed of nitric acid trihydrate (NAT), the sub-visible PSCs form at ambient temperatures well above the ice frost point, but also above the temperature at which supercooled ternary solution (STS) grows out of the background supercooled binary solution (SBS) distribution. The temperature regime of

their formation, inferred from the simultaneous uptake of ambient $HNO_3$ into NAT and their Lagrangian temperature histories, is at a depression of a few kelvin with respect to the NAT existence threshold, $T_{NAT}$. Therefore, their nucleation requires a considerable supersaturation of $HNO_3$ over NAT, and is consistent with a recently described heterogeneous nucleation process on solid foreign nuclei immersed in liquid aerosol. We make a detailed investigation of the comparative limits of detection of PSCs and the resulting sequestration of $HNO_3$ imposed by lidar, mid-infrared and microwave techniques. We find that the

temperature history of air parcels, in addition to the local ambient temperature, is an important factor in the relative frequency of formation of liquid/solid PSCs. We conclude that the initiation of NAT nucleation and the subsequent development of large NAT particles capable of sedimentation and denitrification in the early winter does not emanate from an ice-seeding process. Finally, we investigate the patterns of interannual variability and compare the relative formation frequency of liquid and solid PSCs in the Antarctic lower polar stratosphere using the results of a cluster analysis to synthesize the combined CALIOP and

MLS measurements into a relatively small number of interrelated categories.

## 1 Introduction

In Lambert et al. (2012) (hereafter L12) we reported on the formation of PSCs and the initial stages of denitrification in the early 2008 Antarctic PSC season. The first appearance of PSCs was observed through the uptake of gas-phase $HNO_3$ by MLS and by patchy lidar backscatter detection by CALIOP. Although the uptake of $HNO_3$ was substantial, the low lidar backscatter

and ambient temperatures indicated a nitric acid trihydrate (NAT) composition rather than supercooled ternary solution (STS).





An inference of large particle sizes with radii $\geq$ 5-7 µm and low NAT number density ($< 1 \times 10^{-3}\,\text{cm}^{-3}$) was made from the combination of the temperature, $HNO_3$ and backscatter data, further confirmed by the timescale of the appearance of an enhanced $HNO_3$ layer at 68 hPa, caused by sedimentation of NAT from above and evaporation back into the gas phase. We also determined that in 2008 a NAT polar freezing belt (Tabazadeh et al., 2001; Höpfner et al., 2006), generated by gravity-wave

induced ice-seeding, occurred after the first appearance of large particle NAT clouds; i.e. the latter were not causally linked to the mountain wave seeding events. In L12 and Pitts et al. (2013) (hereafter P13), we studied the uptake of $HNO_3$ by different types of PSCs classified by CALIOP as a function of temperature. We showed that the distributions of gas-phase $HNO_3$ vs temperature combined with the independent CALIOP PSC classification (Pitts et al., 2009) provide valuable insights into the PSC formation process. Liquid STS particles exhibit well-defined equilibrium properties, whereas the liquid/solid particle

STS/NAT mixtures exhibit non-equilibrium properties resulting from kinetically limited growth. In this paper we expand our previous investigations to include the distribution of the lidar backscatter of different PSC types as a function of temperature and $HNO_3$ uptake. We also report on the interannual variability in the Antarctic PSC seasons from 2006–2015.

The discrimination between different PSC types at temperatures above the ice frost-point, $T_{\text{ICE}}$, stemming either from the growth of STS on the background liquid supercooled binary solution (SBS) or the nucleation of NAT, provides critical obser-

vations enabling validation of theoretical PSC formation pathways. In the case of NAT, the nucleation processes are still not understood in detail. Whether homogeneous or heterogeneous nucleation is in force, it is the nucleated NAT number density that provides the key to the subsequent microphysical development of the NAT clouds, since rapid nucleation at high super-saturations leads to higher NAT number densities with small particle radii, whereas slow nucleation at low supersaturations produces low NAT number densities and allows the particles to grow to much larger sizes (Jensen et al., 2002). The homoge-

neous nucleation of NAT from STS and the production of large-size NAT in the 2010/2011 Arctic winter has been simulated in the SD-WACCM/CARMA PSC model (Whole-Atmosphere Community Climate Model with Specified Dynamics with the Community Aerosol and Radiation Model for Atmospheres) as described by Zhu et al. (2015). In this model, homogeneous nucleation rates were determined using the nucleation equations derived from laboratory experiments by Tabazadeh et al. (2002), with the free energy tuned by less than 10%. The same nucleation rates were found by Zhu et al. (submitted, 2016) to reproduce

the observed timing of PSC formation during the Antarctic winter of 2011. In contrast, Hoyle et al. (2013) used an extension of the Zurich Optical and Microphysical box Model (ZOMM) (Luo et al., 2003) to include a new pathway of heterogeneous formation of ice and NAT on solid foreign nuclei inclusions, originating from meteoritic dust, that are assumed to be present in at least 50% of all aerosol drops (Curtius et al., 2005).

Hoyle et al. determined that NAT can form heterogeneously at some considerable vapor supersaturation, at temperatures

well above the ice frost-point, on the solid foreign nuclei immersed in STS. Only a limited number of surface inhomogeneities on the foreign nuclei provide favorable active sites such that the NAT nucleation barrier is depressed sufficiently for nucleation to occur. Once the most efficient active sites have caused nucleation at a particular supersaturation, the remaining population of STS/foreign particles have lower quality active sites that require either a higher supersaturation to increase the nucleation rate or waiting for a longer period of time for nucleation at the same supersaturation to occur. Three tuning parameters are used

to control the heterogeneous nucleation rate in the ZOMM model: nucleation barrier, nucleation strength, and compatibility





factor. These were adjusted with consideration of the detection thresholds applied to the model results (Engel et al., 2013; Hoyle et al., 2013), to replicate successfully the CALIOP backscatter observations for a representative orbit. The tuned model was then used to facilitate intercomparisons with the CALIOP observations during December 2009 and to verify the model results for a few case studies in the Arctic (Engel et al., 2013; Hoyle et al., 2013). However, the low visibility of the NAT PSCs,

as we indicated in L12, poses a detection challenge for lidar backscatter techniques, even though the accompanying $HNO_3$ sequestration in NAT can be substantial and detectable as a decrease in the gas-phase $HNO_3$.

In situ measurements of NAT particles on a synoptic scale, with a large-particle mode of around 15 μm diameter, from the Forward Scattering Spectrometer Probe (FSSP) (Molleker et al., 2014), and observations of their attendant denitrification (Woiwode et al., 2014) were reported during the Arctic RECONCILE (reconciliation of essential process parameters for an

enhanced predictibility of Arctic stratospheric ozone) campaign (von Hobe et al., 2013). In order to resolve discrepancies involving implied condensed $HNO_3$ above that available from the gas phase (including additional $HNO_3$ brought down by renitrification) and growth/sedimentation rates that are not commensurate with back trajectories, those authors hypothesized that the particles are non-compact or highly aspherical or NAT-coated ice. Grooß et al. (2014) coupled a saturation-dependent parameterization of the ZOMM heterogeneous NAT scheme with the Chemical Lagrangian Model of the Stratosphere (CLaMS)

and determined that derived PSC properties were in better agreement with CALIOP data than for simulations using a constant rate NAT nucleation, thus confirming the results of Hoyle et al. (2013). Furthermore, Grooß et al. (2014) compared their simulations to the Arctic in situ aerosol size distributions from FSSP in January 2010 and found that the observed large NAT particles (exceeding 15 μm diameter) were not present in the co-located simulations. However, the CLaMS simulations did develop particle distributions with median NAT diameters of up to 20 μm at other times in the 2009/2010 Arctic winter. Hence,

the new ZOMM nucleation scheme is quite capable of producing large-size NAT, and Grooß et al. speculate that the presence of a highly non-spherical NAT component coupled with the orientation of the particles within the FSSP sample volume may lead to an overestimation of their actual size. Zhu et al. (2015) also found that particles with diameters of 20 μm were produced with their homogeneous nucleation scheme, which does not require that there are special nuclei only available under certain conditions.

PSC formation processes at play in the Arctic are clearly applicable to the Antarctic, and the recent observations from RECONCILE add support to the conclusions of our previous work (L12), which highlighted the appearance of synoptic-scale large-particle NAT in the early Antarctic winter of 2008. Obviously, the hypotheses concerning the origin and microphysical characteristics of these so-called "NAT rocks" will be a challenge to validate without further in situ observations. However, the decade-long record of overlapping spaceborne CALIOP and MLS measurements presents an opportunity to develop improved

algorithms for the extraction of information on PSCs and to apply new-found knowledge to the understanding of their current and future role in ozone depletion.

In section 2 we review the satellite instruments and atmospheric measurements used in our analyses. The temperature history of an air parcel and the relation to heterogeneous nucleation of NAT is explored. We introduce a compact alternative visualization to the standard graphical representation of satellite orbit plots that enables easier comprehension of several parameters

plotted at multiple atmospheric levels and spanning many days of observations. In section 3 we investigate the limits of detec-



tion of equilibrium STS and STS/NAT mixtures separately for lidar backscatter, mid-infrared extinction and uptake of $HNO_3$ from the gas phase. We also investigate STS and NAT PSCs in terms of the distributions in a 3-parameter space of $HNO_3$, backscatter and temperature. We compare these with the CALIOP PSC classification scheme, which uses fixed regions within a 2-parameter discrimination domain (depolarization vs total backscatter). In section 4 we show orbit transects and time-series of colocated CALIOP and MLS data. These are used to investigate the relative formation of liquid/solid PSCs and the resulting denitrification and renitrification. In section 5 the early stages of formation of Antarctic PSCs at 68–21 hPa in 2009 are examined using CALIOP PSC types and Lagrangian temperature history, with the inference of an initial population of sub-visible solid-particle NAT clouds superceded by a predominantly liquid STS composition over a period of about a week. Finally, the interannual variability of the early Antarctic PSC seasons from 2006–2015 is discussed in section 6.

## 2 Datasets and methodology

The Cloud-Aerosol Lidar with Orthogonal Polarization (CALIOP) dual-wavelength elastic backscatter lidar (Winker et al., 2009) flies on the Cloud-Aerosol Lidar and Infrared Pathfinder Satellite Observations (CALIPSO) satellite launched in April 2006. The Microwave Limb Sounder (MLS) is onboard the Aura spacecraft launched in July 2004. CALIPSO and Aura are part of the NASA/ESA afternoon "A-train" satellite constellation at 705 km nominal altitude and 98° inclination, with daily global coverage attained in 14.5 orbits. The initial A-train configuration of the CALIPSO and Aura spacecraft from April 2006 to April 2008 resulted in an across-track orbit offset of ~200 km, with the MLS tangent point leading the CALIOP nadir view by about 7.5 minutes. Since April 2008 Aura and CALIPSO have been operated to maintain positioning within tightly constrained control boxes, such that the MLS tangent point and the CALIOP nadir view are colocated to better than about 10-20 km and about 30 seconds.

We derive collocated meteorological data from the Goddard Earth Observing System Data Assimilation System (GEOS-5 DAS). The 6-hourly synoptic gridded data products (Rienecker et al., 2008) of temperatures and winds are supplied on a 540 by 361 longitude-latitude grid. The GEOS-5 data are interpolated in location and time to the MLS along-track data. Parcel temperature histories are obtained from the MLS Lagrangian Trajectory Diagnostic (LTD) dataset (Livesey et al., 2015), which consists of 15-day forward and reverse trajectories launched from a curtain of points along the Aura MLS observation track. The advection calculations are based on Manney et al. (1994), with wind fields and diabatic heating rates taken from the 3-hourly Modern-Era Retrospective Analysis for Research and Applications (MERRA-2) dataset (Bosilovich et al., 2015). Advances in both the GEOS-5 model and the assimilation system, including GPS Radio Occultation datasets, are included in MERRA-2. The integration uses a fourth-order Runge–Kutta scheme with a 5 minute time-step, and saved trajectory locations and temperatures are output every 30 minutes. The MLS Derived Meteorological Products (DMPs) (Manney et al., 2007) are used where necessary to identify measurement locations that lie within the Antarctic vortex based on scaled potential vorticity, $sPV < -1.4$.



## 2.1 CALIOP PSC data

We use the CALIOP Level-1b v3 standard data product to extract information on PSCs (as documented in L12) at a 50 km horizontal by 0.5 km vertical resolution. We also use a recently released Level 2 operational dataset L2PSCMask (v1 Polar Stratospheric Cloud Mask Product) produced by the CALIPSO science team. The Level-2 operational data consist of nighttime only data and contain profiles of PSC presence, composition, optical properties, and meteorological information along the CALIPSO orbit tracks at 5 km horizontal by 180 m vertical resolution. We have determined that the v1 L2PSCMask operational product has an incorrect separation of the `MIX1` and `MIX2` classifications and does not follow the boundary specification given by Pitts et al. (2009). The following three-step algorithm (using the Scientific Data Set variable names supplied with the CALIPSO Hierarchical Data Files (HDF) files) has been applied to generate the correct `MIX1` and `MIX2` classes:

```
INVBR = 1. − 1./TOTAL_SCATTERING_RATIO_532
MIX1 = PSC_Composition EQ 2
       OR (PSC_Composition EQ 3 AND (INVBR LE 0.2))
MIX2 = PSC_Composition EQ 3 AND (INVBR GT 0.2)
```

## 2.2 MLS gas-phase constituents

The Microwave Limb Sounder measures thermal emission at millimeter and sub-millimeter wavelengths from the Earth's limb (Waters et al., 2006) along the forward direction of the Aura spacecraft flight track, with a vertical scan from the surface to 90 km every 24.7 s. Each orbit consists of 240 scans spaced at $1.5°$ (165 km) along-track, with a total of almost 3500 profiles per day and a latitudinal coverage of $82°$ S to $82°$ N. The Level-1 limb radiance measurements are inverted using 2-D optimal estimation (Livesey et al., 2006) to produce Level-2 profiles of atmospheric temperature and composition. Here we use the MLS version 4 (v4) data (Livesey et al., 2016) with single-profile precisions of 4–15 % for $H_2O$ (Read et al., 2007; Lambert et al., 2007) and 0.7 ppbv for $HNO_3$ (Santee et al., 2007).

## 2.3 Temperature history and relation to NAT nucleation and growth processes

In this work we frequently apply a convenient temperature coordinate transformation, $T - T_{ICE}$, by using MLS $H_2O$ to calculate the ice frost point, in order to remove height-related variations due to changes in the $H_2O$ partial pressure (L12 and P13). As in L12 we quantify the duration of exposure of an air parcel to low temperatures by defining the temperature threshold exposure (TTE) as the total integrated time the air parcel is subject to synoptic-scale temperatures below the chosen threshold. We use a threshold of $T_{ICE} + 4$ K (aproximately $T_{NAT} - 3.5$ K) to demonstrate empirically the correlation of TTE with the uptake of $HNO_3$ by NAT PSCs. The temperature history follows a diabatic back-trajectory for up to 15 days obtained from the MLS LTD dataset (Livesey et al., 2015). The TTE is the total time (in days) that an air parcel has been exposed to temperatures below $T_{ICE} + 4$ K since the last time the temperature fell below $T_{NAT}$ and remained below $T_{NAT}$ consistently, i.e. any number





of episodes of cooling below $T_{ICE} + 4$ K are accumulated providing that the air parcel has remained consistently below $T_{NAT}$. The $HNO_3$ and $H_2O$ values (for estimating $T_{NAT}$ and $T_{ICE}$) are assumed to be the same at all back-trajectory times. Typical lower stratospheric polar values for $T_{ICE}$ and $T_{NAT}$ are 188 K and 195 K, respectively. TTE is a remarkably good indicator of the geographical extent of the $HNO_3$ depletion in the vortex (L12). Here, we explore the correspondence of TTE with NAT

nucleation and growth processes.

According to the development of NAT along a sample trajectory shown by Hoyle et al. (2013) in their Figure 1, substantial nucleation begins only for temperatures below $T_{NAT} - 4$ K. We investigate the temperature-time domain of the nucleation process in Figure 1, for both (a) early season unperturbed and (b) late season denitrified atmospheres, and calculate the resulting NAT number densities using the heterogeneous nucleation scheme given in Hoyle et al. The figure serves to illustrate the general

properties of NAT cloud formation, but in reality the temperature-time path is important for the modeling of specific clouds. The nucleation rate is a strong function of temperature, and the nucleated NAT shows an almost step-like transition over a narrow temperature range. In contrast, the variation of NAT density with exposure time is more gradual because the nucleation rate at a fixed temperature (i.e. fixed supersaturation) depends only on the integration over time. Exposure to temperatures of $T_{NAT} - 2.2$ K and above produces negligible NAT densities ($< 10^{-6}$ cm$^{-3}$), even for time durations exceeding a month,

whereas exposures of $\sim$ one day to temperatures near $T_{NAT} - 4$ K produces NAT densities three orders of magnitude greater ($\sim 10^{-3}$ cm$^{-3}$). At lower temperatures, the NAT saturation ratio is limited by uptake of $HNO_3$ from the gas phase by STS (Luo et al., 2003), causing the curvature of the NAT density contours below $T_{NAT} - 4.5$ K. Denitrification has little effect on the sharp temperature transition, but at lower temperatures where STS forms there is a visible decrease in the NAT number density generated for the same time duration. At nucleation, the NAT particle sizes are small and no larger than the progenitor

aerosol particle (SBS or STS). Hence, considerable growth of the NAT particles is required before they can be detected using remote sensing techniques. TTE can be viewed as a proxy for the time elapsed since nucleation occurred, i.e. as a measure of the effective growth-time of the NAT particles. The NAT volume density increases gradually through sequestration of $HNO_3$ from the ambient gas phase, and the particles may not achieve their much larger equilibrium size until several days following nucleation.

## 25  2.4  Visual representation of satellite orbital data

Figure 2a shows typical orbit tracks of the MLS $HNO_3$ distribution mapped onto a polar stereographic projection for ascending and descending orbits at 32 hPa. Each colored square is centered at the corresponding MLS profile latitude/longitude retrieval location. The along-track spacing is $1.5°$ (distance between centers of the squares). Note that the dimensions of the squares are not related to the MLS along-track (several hundred km for $HNO_3$) or across-track (about 10 km) resolutions. Orbit numbering

(0 is the day start orbit) is shown around the $60°$ S latitude circle. The color scheme indicates contrasting colors either side of 10 ppbv. Over the 10-day period (26 May to 5 June 2009, Figure 2a and b), the $HNO_3$ values decrease and the area with values $< 10$ ppbv is seen to increase and move eastward. Such "satellite data orbit plots" are commonly used, but they do not scale up easily for viewing multiple species and pressure levels over periods of the order of a month; e.g. to track the evolution of four parameters over twenty days at four distinct atmospheric levels requires the digestion of 320 images.



We present an alternative scheme, designed to improve the data visualization, with some similarities to the familiar Hovmöller diagram, but with the abscissa following selected sections of the satellite orbit track rather than running along a zonal or meridional circle. Figure 2c shows the same data as in Figure 2a, but replotted as a time-ordered sequence of the along-track points. The MLS orbit tracks are unfolded along the abscissa as a function of the along-track angle ($1.5°$ is the angular spacing),

where the along-track angle of $30°$ corresponds to the closest approach of the orbit to the South Pole (at about latitude $82°$ S). Also shown for convenience are the along-track distances (in km), and the corresponding latitudes and solar zenith angles. The MLS measurement time (hours since start of day at 0UT) is on the ordinate. The orbit numbers are given next to the right hand ordinate. Again the dimensions of the squares are not related to the MLS orbital track spacing or the MLS measurement time (each complete vertical atmospheric profile is accumulated over about 25 s). The main purpose of this compact visual represen-

tation (i.e. a sparse "raster" image) is that it enables the "raw" daily observations to be stacked into a longer time-series without involving gridding onto a map projection. Note that there is a geographical data void (within $8°$ of the pole, see Figure 2a and b) that is not apparent in the raster representation. MLS looks forward in the along-track direction, so it never actually looks into the $8°$ polar cap, nor does CALIOP with its nadir view. While the low $HNO_3$ in the vortex rotates eastward in the orbit track plot, this motion translates into a time displacement in the raster plot. The ascending/descending tracks do not intersect

in the raster plot and the terminator is always on the rightmost side (a feature that could be potentially useful for examining diurnal species such as ClO).

## 3    Detection and classification of PSCs

Improvements in modeling capabilities drive a commensurate need for a thorough evaluation of the observational aspects of PSC research, such as biases in temperature analyses, derived air parcel temperature histories along trajectories and instrument

measurement biases and uncertainties. This is highlighted in a specific example shown by Hoyle et al. (2013) in the Arctic on 26 December 2009 (their Figure 7, orbit 26_04), where they indicate that the ZOMM model predicts a secondary area of NAT clouds between longitudes $12°–53°$, whereas the CALIOP observations do not show a corresponding lidar detection. Hoyle et al. noted the prolonged exposure time (80h) resulting in sedimentation of NAT as a potential cause of the lack of coincident lidar signal from CALIOP in the L2PSCMask product. However, we have determined that smoothing of the

CALIOP L2PSCMask data using a 5x5 median filter to improve signal to noise does in fact indicate the presence of PSCs. In Figure 3, we compare the along-track cross-sections of the MLS $HNO_3$, CALIOP L2PSCMask, and the smoothed total and perpendicular backscatter ratios. Additionally, inspection of the MLS gas-phase $HNO_3$ identifies a coincident decrease also consistent with the location of the CALIOP PSCs (note that the ZOMM model $HNO_3$ is not shown by Hoyle et al.).

### 3.1    Modeled uptake of $HNO_3$, lidar backscatter and infrared extinction in PSCs

We model the microphysics of representative STS and NAT particle distributions according to the methodology given in Pitts et al. (2009) and L12. For the lidar scattering calculations, Mie theory is used for liquid spherical particles and the T-matrix (Mishchenko and Travis, 1998) for solid NAT particles. As we noted in L12, the NAT particle shape is an open issue, and we



continue here to use a range of spheroidal shapes to illustrate the lidar sensitivity to NAT. Real refractive indices at 532 nm were assumed to be 1.43 for STS and 1.50 for NAT, with zero imaginary refractive indices for both particle types. For the mid-infrared region, complex refractive indices were obtained from the tabulations given by Myhre et al. (2005) for STS and Toon et al. (1994) for NAT.

Existence temperatures of the PSC types are calculated using equilibrium thermodynamics and are dependent on the ambient partial pressures of $H_2O$ in the case of the ice frost point, $T_{ICE}$ (Murphy and Koop, 2005), and also $HNO_3$ for $T_{NAT}$ (Hanson and Mauersberger, 1988) and STS (Carlsaw et al., 1995). SBS particles grow by condensation on cooling, first by uptake of $H_2SO_4$ and $H_2O$ from the gas phase and then, at sufficiently low temperatures, uptake of $HNO_3$ occurs, forming STS at a few kelvin below the NAT point close to $T_{STS} \sim T_{NAT} - 3.5\,K$ (Carlsaw et al., 1997; Drdla et al., 2003). In the polar stratosphere,

NAT is thermodynamically stable at temperatures below $T_{NAT} \sim T_{ICE} + 7\,K$, although the NAT nucleation process is still not understood in detail. The PSC detection limits for lidar backscatter and infrared extinction are dependent on the background aerosol loading in addition to measurement noise. Under conditions of high quiescent background aerosol loadings (e.g. at times perturbed by volcanic aerosol), a higher threshold is required to discriminate PSCs from the background. The thresholds also depend on vertical and horizontal averaging. For illustration here we use typical threshold levels appropriate for the

CALIOP lidar with a 50 km by 0.5 km resolution (L12): total backscatter ratio, $R_T = 1.25$, and perpendicular backscatter threshold, $\beta_\perp = 2.5 \times 10^{-6}\,km^{-1}\,sr^{-1}$. For a mid-infrared limb sounder operating in the window-region near 12 μm we use an extinction threshold, $k_{ext} = 4 \times 10^{-5}\,km^{-1}$ based on measurements by the Improved Stratosphere and Mesosphere Sounder (Lambert et al., 1996). The threshold level for detection of the gas-phase removal resulting from the ambient $HNO_3$ uptake in STS is taken to be 1 ppbv, slightly higher than the 0.7 ppbv MLS measurement uncertainty.

**3.1.1   Equilibrium STS**

Figure 4a shows the temperature variation of the modeled equilibrium thermodynamic properties of STS based on the Carlsaw et al. (1995) parameterization and the calculated lidar backscatter and 12 μm infrared extinction assuming Mie theory. The calculations assume a pressure of 46 hPa, 5 ppmv $H_2O$, and 12 ppbv total $HNO_3$. In situ observations of the STS aerosol volume are well modeled (Carlsaw et al., 1994), as is the uptake of $HNO_3$ from the gas phase (L12). Uptake of $HNO_3$ from

the gas phase varies rapidly (50, 10, 1 %) over a narrow temperature range ($T - T_{ICE} = 2.3, 3.1, 3.9\,K$), and the maximum temperature derivative ($-6.7$ ppbv / K) is at $T - T_{ICE} = 2.9\,K$. The theoretical residual gas-phase $HNO_3$, accounting for the uptake in STS, is shown in Figure 4b as a function of the total backscatter and 12 μm infrared extinction. A scatter plot of the observations of $HNO_3$ and backscatter (or extinction) in the presence of STS are expected to lie beneath the theoretical curve, and this is investigated in Section 3.2. Figure 4c shows that the infrared extinction is more sensitive to STS than the lidar

backscatter, since the corresponding threshold equivalent condensed $HNO_3$ contents of STS are 0.5 and 0.8 ppbv for the two measurement approaches, respectively.





### 3.1.2 STS/NAT mixtures

The morphology of NAT particles is still an open question, as is the compactness of the particles (Molleker et al., 2014; Woiwode et al., 2014). Light scattering studies have repeatedly shown that detailed particle morphology cannot be deduced from the depolarization; e.g. Nousiainen et al. (2012) investigated simple and complex shapes with size parameters (ratio of the
particle circumference to the wavelength) in the range 2–12 and real refractive indices in the range 1.55–1.603 representative of silicate particles and noted similar depolarization ranges for the fifteen different shapes (regular and irregular) that were analysed. Therefore, by analogy, the selection of a few aspect ratios for a simple spheroidal shape is sufficient to demonstrate the variations in lidar backscatter properties and to highlight the challenges for lidar measurements to detect low number density large particle radii NAT. The NAT particles are modeled as spheroids with diameter-to-lengths (aspect ratios), $\epsilon$, both
oblate (1.2) and prolate (0.8, 0.9, 0.95) using T-matrix calculations assuming a power-law NAT size distribution; otherwise the STS/NAT mixtures are modeled as in Pitts et al. (2009) and L12. Previous investigations (Liu and Mishchenko, 2001; Flentje et al., 2002, L12) have noted that larger depolarizations (over 60%) result from the more nearly spherical particles in the aspect ratio range $\epsilon = 0.90 - 1.10$. Recent analyses of the CALIOP data (Engel et al., 2013; Hoyle et al., 2013) have used $\epsilon = 0.90$ for NAT to improve modeling of the observed CALIOP depolarization range. Note that although the total backscatter is often
dominated by STS, the use of a perpendicular backscatter threshold (L12, P13) rather than an aerosol depolarization threshold reduces the possibility of the STS signal to mask the NAT signal in STS/NAT mixtures (Biele et al., 2001).

     The large-particle example shown in Figure 5a is for a NAT particle distribution with a number density $N_{\mathrm{NAT}} = 0.001 \, \mathrm{cm}^{-3}$ and effective radius $R_{\mathrm{eff}} = 6.5 \, \mu\mathrm{m}$. Note that in this example the uptake of $HNO_3$ follows the NAT equilibrium curve until the saturation point is reached and the condensed $HNO_3$ equals the volume in the assumed NAT particle distribution (plateau
region with 4 ppbv condensed $HNO_3$). No further uptake of $HNO_3$ occurs until the temperature decreases sufficiently to allow growth of STS.

     The lidar detection of NAT (back scattering in visible spectrum) is sensitive to the asphericity parameter, whereas the infrared extinction (dependent mainly on emission and therefore particle volume) is not. Hence, there are four lidar total backscatter curves corresponding to each aspect ratio in Figure 5a, but only a single infrared extinction curve is plotted representative of all
four. Likewise, the inferred uptake of $HNO_3$ by STS/NAT as measured by microwave observations is also insensitive to particle shape (in addition, the aerosol emission is negligible in the microwave region and has no effect on the gas measurements). In the absence of an actual PSC detection, the observed reduction in gas-phase $HNO_3$ is an indirect detection method of PSC activity (L12), since either the $HNO_3$ is condensed into NAT PSCs below the detection threshold of the lidar or the NAT particles were so large they sedimented and the missing gas-phase $HNO_3$ is the result of permanent denitrification.
The theoretical condensed $HNO_3$ in NAT at a fixed temperature of $T - T_{\mathrm{ICE}} = 5$ K is shown in Figure 5b as a function of the NAT number density. At this temperature, the STS contribution to the backscatter, infrared extinction and $HNO_3$ uptake is negligible (see Figure 4) and we may safely concentrate on the properties of the NAT particles alone. The temperature is about 2.5 K below the NAT existence temperature and about 2 K above the temperature at which substantial uptake of $HNO_3$ into STS occurs. The red diamond symbol marks the detection threshold for 1 ppbv of $HNO_3$ condensed in NAT (or





equivalently a 1 ppbv uptake of $HNO_3$ from the gas phase) and corresponds to a NAT number density of $2.4 \times 10^{-4}$ cm$^{-3}$. In Figure 5c we show the total lidar backscatter and infrared extinction as a function of the NAT number density. The total backscatter detection threshold (1.25) for the four particle aspect ratios corresponds to NAT number densities ranging from 4.9 to $8.4 \times 10^{-4}$ cm$^{-3}$ compared to $1.1 \times 10^{-4}$ cm$^{-3}$ for the infrared extinction threshold ($4 \times 10^{-5}$ km$^{-1}$). For this large particle

radius example, uptake of $HNO_3$ and infrared detections are more sensitive than the lidar total backscatter. The perpendicular backscatter coefficient (Figure 5d) is more sensitive than the total backscatter to the presence of non-spherical NAT (resulting in the detection of lower number densities ranging from 1.2 to $3.6 \times 10^{-4}$ cm$^{-3}$, except for the 0.8 aspect ratio, which shows less sensitivity. Hence, operation of a lidar with an orthogonal polarization channel can substantially improve the detection threshold for non-spherical NAT for some aspect ratios. However, the infrared detection is still shown to be more sensitive for

the large particle range.

### 3.1.3 Intercomparisons of PSC detection techniques

The sensitivity of different techniques employed to detect PSCs over a wide range of number densities and effective radii can be illustrated with Figure 6. Selected detection limits for CALIOP lidar, infrared limb emission and inferred detection by the measured uptake of $HNO_3$ from the gas phase or the in situ aerosol detection of condensed $HNO_3$ in NAT are shown. Again, the

15 calculations are for a temperature $T - T_{ICE} = 5$ K, since the purpose of the comparison is to show the potentially large variation in lidar backscatter response to NAT PSCs of differing asphericities. The lines in Figure 6 mark the detection limits for the various techniques and indicate the lowest NAT number density ($N_{NAT}$) that can be detected for a given NAT effective radius ($R_{eff}$); i.e. any combination of ($N_{NAT}, R_{eff}$) lying below a given line is below the detection limit for that particular technique. The gray shaded region indicates the region that is below the detection threshold for any of the techniques assuming the given

variation in NAT aspect ratio. The solid (dashed) purple-blue lines correspond to the detection thresholds for total backscatter (perpendicular backscatter) for different aspect ratios. As expected from the previous section, the perpendicular backscatter is in general more sensitive to the presence of low number density NAT than is the total backscatter ratio. However, there is substantial variation in the sensitivity to the particle aspect ratio, $\epsilon$. The sensitivity curves for infrared extinction (yellow) and lidar perpendicular backscatter are similar for the $\epsilon = 0.95$ case, whereas for $\epsilon = 0.8$ the lidar is much less sensitive.

For low number density / large particle NAT (bottom-right of Figure 6), the uptake of $HNO_3$ from the gas phase (red) can still be quite substantial (1 ppbv) and is independent of the particle asphericity. The extremes of the lidar sensitivity to small NAT ($R_{eff} < 1$ μm) (ranging over an order of magnitude in particle number density) are seen to be reversed for large NAT ($R_{eff} > 5$ μm); i.e. the lidar technique is more sensitive to small NAT with an aspect ratio of 0.80 than 0.95 and vice versa for large NAT. Except for the aspect ratio of 0.95, the detection limits for infrared emission and the uptake of 1 ppbv $HNO_3$ from

the gas phase become more sensitive than the lidar limit for large NAT with $R_{eff} > 2.5$ and $R_{eff} > 5$ μm, respectively. The near coincidence of the green dashed line and red solid line shows that a NAT volume density of 0.2 μm$^3$ cm$^{-3}$ is approximately equivalent to 1 ppbv of condensed $HNO_3$.





### 3.2 Separation of PSC types using backscatter, HNO₃ and temperature

In this section we investigate the lidar PSC classification by exploring the 2-dimensional cross-sections resulting from projections of the 3-dimensional coordinate space of temperature, $HNO_3$ and total backscatter. Here we use the CALIOP Level-1b v3 data with coincident MLS data processed as detailed in L12 for the period 10 May to 25 October 2009 in the Antarctic at 46–21 hPa. In Figure 7 we show probability density functions (PDFs) classified according to the CALIOP PSC scheme (Pitts et al., 2009), with modifications discussed in L12 and P13, in six columns ( `ALL`, `LIQ`, `MIX1`, `MIX2`, `ICE` and `None`) and four rows (described below). The `ALL` class is the sum of the individually classified PSC components `LIQ`, `MIX1`, `MIX2` and `ICE`. The `None` class represents all cases below the CALIOP detection threshold. The four rows for each column show the depolarization vs normalized backscatter and the corresponding PDFs for the three possible combinations of pairings from the temperature, $HNO_3$ and total backscatter coordinates in the PSC classification. Nitric acid vs temperature has been shown previously by L12 and P13 (and is included for completeness), but backscatter vs temperature and $HNO_3$ vs backscatter are shown for the first time here.

**Row 1:** Depolarization ($\delta$) vs normalized backscatter ($1 - 1/R_T$):

CALIOP data analysis, detection and classification are discussed in L12 and P13. The PSC classes are shown in the CALIOP depolarization vs backscatter classification diagram in the first row. Black solid lines indicate the main PSC types. Class boundaries for `MIX2-enh` and wave ice ($R_T > 50$) are shown as black dashed lines, but are not differentiated here from the `MIX2` and `ICE` main classes. The classification boundaries were originally chosen (Pitts et al., 2009) to distinguish STS (depolarization less than 3%), STS/NAT mixtures (significant depolarization indicating a solid component) and ice. Note that the `LIQ`/`MIX1` class boundary is fuzzy, and depolarization values (the ratio of the perpendicular to parallel backscatter) can exceed 3% for `LIQ` because of measurement noise even though the perpendicular backscatter component indicates below-threshold response. Similarly, the `None` class boundary is fuzzy because of measurement noise. The black-white dashed line shows the theoretical lower limit of detection as a locus of points ($\delta$, $1 - 1/R_T$) for the chosen perpendicular backscatter threshold, $\beta_\perp = 2.5 \times 10^{-6}$ km$^{-1}$ sr$^{-1}$, calculated for a typical polar atmosphere from the expression

$$R_T(\delta) = \left[ \left( 1 + \frac{1}{\delta} \right) (\beta_\perp - \beta_{m_\perp}) + \beta_m^T \right] \frac{1}{\beta_m^T}$$

where the molecular depolarization $\delta_m$ is 0.0036, the molecular perpendicular backscatter component, $\beta_{m_\perp}$ is $\beta_m^T \frac{\delta_m}{1+\delta_m}$, the total Rayleigh scattering (both polarizations) is $\beta_m^T$, and the fractional depolarization range is $\delta = 0 \ldots 1$. This low detection limit is not strictly attained in practice because of additional spatial coherence constraints that are imposed to reduce false positives to less than 0.1% (L12). The coherence constraint results in the distribution of points in the `None` class appearing to the right side of the black-white dashed line. All the other imposed class boundaries are sharp, although this does not imply that the distinction between these PSC types is definitive. For example, as noted in P13, the `ICE` "arm" close to the `LIQ` class (normalized backscatter 0.7–0.85) is intersected by the `MIX2`/`ICE` boundary. Better separation between the `MIX2` and `ICE` classes based on allowing for the seasonal variation in the location of the ice "arm" associated with denitrification was discussed in P13. Overall, the classification using the 2-D regions of the depolarization vs normalized backscatter provides very





good discrimination between STS and solid-particle PSCs. Separation of PSCs into the `MIX1` and `MIX2` classes is somewhat arbitrary, but provides useful information on the NAT number densities (Pitts et al., 2009).

**Row 2:** Gas-phase $HNO_3$ vs $T - T_{ICE}$:

In L12 and P13, we demonstrated that the uptake of $HNO_3$ vs temperature, classified according to CALIOP PSC types,
is in good agreement with expected temperature existence regimes for STS, NAT and ice, except for an apparent $\sim$1 K bias with respect to the GEOS-5 temperatures. This is consistent with comparisons (not shown) of coincident independent temperatures obtained from the Constellation Observing System for Meteorology, Ionosphere, and Climate (COSMIC) GPS Radio Occultation data, which indicate up to a 0.7 K cold bias in GEOS-5 in the 2009 Antarctic lower stratosphere. The `LIQ` class follows the theoretical equilibrium uptake of $HNO_3$ by STS (blue lines from 2 to 24 ppbv in 2 ppbv steps), and the `MIX1`
and `MIX2` classes show significant non-equilibrium behaviour, along with two discernible branches following the STS and the NAT (green line) equilibrium curves. The `ICE` class is a compact and roughly symmetric distribution, located at the lowest temperatures and lowest $HNO_3$ gas-phase amounts. The leading edge of the $HNO_3$ gas-phase distribution for the `None` class follows the STS uptake curve. Also visible is a separate highly denitrified branch ($HNO_3 < 5$ ppbv) extending to beyond 10 K above the ice frost point.

**Row 3:** Total backscatter ($R_T$) vs $T - T_{ICE}$:

This row shows the temperature domains coreponding to the various CALIOP PSC classes (the region of highest backscatter, $R_T > 10$, is not shown). The blue lines indicate the theoretical STS backscatter vs $T - T_{ICE}$ for gas-phase $HNO_3$ increasing from 2 to 24 ppbv in 2 ppbv steps. Note that the current depolarization/backscatter classification scheme does not use temperature as a discriminant. Backscatter vs temperature is only used in the CALIOP classification scheme to determine daily
detection thresholds (Pitts et al., 2009). The `LIQ` class shows a rapid increase in total backscatter near $T - T_{ICE} = 3.5$ K (i.e. $T_{STS}$ is located at point the blue curves join the abscissa). There is a thin tail which does not reach out as far as $T_{NAT}$ (located at the intersection of the two red-black dashed curves). The envelope of the theoretical STS curves (bounded by total $HNO_3$ values of 16–18 pbbv) is a reasonable description of the distribution, but at the lowest temperatures and highest expected backscatter ($>6$) the region is unpopulated. This may be because such conditions are favorable for freezing of STS to form ice
(Koop et al., 1995). The `MIX1` class shows a tail with very low backscatter extending from $T_{STS}$ out to beyond $T_{NAT}$ and consistent with $N_{NAT} \leq 0.001$ cm$^{-3}$. At temperatures below $T_{STS}$ the backscatter increases, but remains below values of about 3, and the distribution is consistent with $N_{NAT} \leq 0.01$ cm$^{-3}$. The `MIX2` class also shows a tail beyond $T_{NAT}$ and is consistent with the largest $N_{NAT}$ ranging up to slightly above 0.01 cm$^{-3}$, except at the lowest temperatures below $T - T_{ICE} = 1$ K, where the backscatter increases rapidly. The demarcation between the `MIX2` and `ICE` classes at $R_T = 5$ is somewhat arbitrary and
leads to NAT/ice mis-classification in this transition region. This effect is ameliorated by the use of a successive averaging scheme (Pitts et al., 2009), since ice can generally be detected at a higher spatial resolution than is shown here. At the highest backscatter values there is an apparent trend in the `ICE` class towards higher minimum temperatures. The `None` class shows a narrow distribution consistent with the chosen total backscatter threshold of 1.25.

**Row 4:** Gas-phase $HNO_3$ vs total backscatter ($R_T$):



The blue lines indicate the theoretical STS $HNO_3$ vs backscatter curves. These show an almost linear decrease in $HNO_3$ with increasing backscatter. For the LIQ class the regions $R_T <2$ with high $HNO_3$ and $R_T >6$ with low $HNO_3$ are not fully populated when compared to the theoretical curves. For low backscatter this is likely to indicate STS containing less than 0.8 ppbv of $HNO_3$, which cannot be detected. For high backscatter this may suggest the result of freezing of STS to ice (see Row 3 discussion). The LIQ class also shows a bulge in the PDF around $R_T = 4$ to 6 that reaches the theoretical curve for total $HNO_3$=22 ppbv. Since the None class indicates that the maximum $HNO_3$ is 18.5 ppbv, the ∼3.5 ppbv excess $HNO_3$ in the LIQ PDF may have arisen from the formation of additional $HNO_3$ produced from heterogeneous reactions occurring on the liquid particles and released into the gas phase or by renitrification from evaporation of sedimenting NAT clouds (see Section 4.2). The None class also indicates totally denitrified regions (consistent with the noise floor of the MLS measurements) with insufficient $HNO_3$ to form any kind of non-ice PSCs. Note that the data in this row are independent of the suspected GEOS-5 temperature bias.

## 4 Evaluation of CALIOP and MLS colocated measurements

Although simultaneous colocated measurements of PSCs and gas constituents are obviously to be preferred over spatially and temporally decorrelated measurements, the availability of such measurements from MLS and CALIOP cannot be expected to provide full closure to the questions of PSC formation. Careful consideration of the details of PSC formation is required to reconcile the pieces of information garnered from the different measurement techniques. In this regard the ultimate aim would seem to be Lagrangian measurements following the full life-cycle of PSC evolution. However, further unresolved issues have emerged from the long-duration stratospheric balloon flights by Ward et al. (2014), who describe measurements of the NAT nucleation rate that show much larger spatial inhomogeneities in NAT occurrence than anticipated.

### 4.1 MLS and CALIOP orbit transects

We have selected some views from the combined MLS and CALIOP data record to illustrate how the interpretation of the morphology of PSCs and gas-phase $HNO_3$ in along-track transects is governed not only by the local ambient temperature, but also by the underlying temperature histories. Here we use the CALIOP Level 2 v1 PSC Mask dataset and also apply post-processing to generate coarser horizontal/vertical bins for a better comparison at the scale of the MLS along-track and vertical resolution. Each averaging bin is the size of the MLS along-track separation (165 km) and the height between the mid-points of the pressure levels (2.7 km) for the MLS $HNO_3$ data product. Note that in this section we use the L2PSCMask data class name STS instead of LIQ.

In Figure 8 we present along-track orbit transects in the 2009 Antarctic early winter. The four columns show a sample of orbit tracks over an 11-day period (day number / orbit number): 2009d136/11, 2009d145/6, 2009d145/13, 2009d147/9. The data in each row are (a) MLS $HNO_3$, (b) $T - T_{ICE}$, (c) temperature history (TTE), (d) temperatures following the Lagrangian back-trajectories, (e) L2PSCMask CALIOP PSC classification and (f) the post-processed Liquid/Solid index, LS_index = $(L-S)/(L+S)$, where $L$ is the number of observations in the STS classification and $S$ is any other PSC (solid) detection. The



`LS_index` is a ratio of the number of CALIOP classifications occurring within the corresponding MLS along-track extent, with the extreme values of $-1$ indicating only solid class and $+1$ only liquid class detections. The `LS_index` represents the dominant PSC classification in a sample volume similar in size to the MLS gas species resolution. The pixel size is much larger than that in the L2PSCMask composition plot, and the composition speckle can be seen as "blocky" regions in the `LS_index`.

Several contour lines are superposed on the orbit transects: The black/white quasi-horizontal labeled contours show a sample of the MLS pressure levels ($HNO_3$ is retrieved at 6-levels/decade change in pressure). The green and blue contours represent temperatures corresponding to the $T_{NAT}$ threshold (using GEOS-5 and MLS $HNO_3$, $H_2O$) and $T_{ICE} + 2$ K, respectively. The blue temperature contour encompasses an expected $HNO_3$ uptake of about 50% from the gas-phase into STS (see Figure 4a). The yellow contour is the $HNO_3$ 12 ppbv contour, and the red contour encloses the area with TTE $\geq$ 3 days.

**Case 1:** 16 May 2009 / 2009d136 / 11: MLS $HNO_3$ shows an extent of a few hundred km with $HNO_3$ uptake from the gas phase, which is offset from the region of the lowest local temperatures (Figure 8b), but more closely located with peak values of TTE (Figure 8c) (which are still below 3 days). CALIOP detects some pixels of `MIX1` and fewer of `MIX2` (Figure 8e). Another orbit track (9, not shown) on the same day shows smaller $HNO_3$ uptake with no coincident PSC detections.

**Case 2:** 25 May 2009 / 2009d145 / 6: MLS $HNO_3$ shows a substantial region of $HNO_3$ uptake over 1500 km and suggests
a combination of uptake in two separate regions, one located centrally within the local temperature minimum (blue contour, $T_{ICE} + 2$ K) and another offset extending to the right edge of the $T_{NAT}$ contour (green). The temperature history (red contour) is the key to this rather apparent asymmetry of the $HNO_3$ distribution with respect to the $T_{NAT}$ contour, since the TTE clearly has larger values outside of the central local temperature minimum and is associated with the region of $HNO_3$ uptake on the right. The L2PSCMask (Figure 8e) shows a substantial STS cloud with some composition speckle, mainly coincident with
the central local temperature minimum. The `LS_index` (Figure 8f) shows predominantly liquid detections, with more solid detections at the top and lower right edge of the cloud. The STS class (Figure 8e) does not completely fill the local temperature contour on the right-hand side, which overlaps with the largest TTE values (red contour).

**Case 3:** 25 May 2009 / 2009d145 / 13: MLS $HNO_3$ shows significant uptake coincident with the peak temperature exposure history. The L2PSCMask (Figure 8e) shows some `MIX1`/`MIX2` class, but not coincident with the largest $HNO_3$ uptake. A
very small area of local temperature minimum (blue line) at 32 hPa shows little $HNO_3$ uptake and some STS class pixels.

**Case 4:** 27 May 2009 / 2009d147 / 9: MLS $HNO_3$ indicates substantial $HNO_3$ uptake (Figure 8a) coincident with the local temperature minimum (blue contour), but also extending to the left edge of the $T_{NAT}$ region (green contour). The greatest exposure to low temperatures (red contour) is associated with the left region of $HNO_3$ uptake. The L2PSCMask (Figure 8e) shows substantial STS class (with multi-class speckle), but only in the right half of the minimum local temperature region.
The left half is the area with the largest TTE values (red contour). The L2PSC_Mask shows `MIX1`/`MIX2` class pixels below and to the left of the large STS cloud and also in the regions outside of the TTE and $T_{NAT}$ contours. We also note that although the minimum temperatures along a number of the back trajectories passed below the frost point within two days of the MLS/CALIOP observations, no ice PSCs were detected.

Examination of the overlaps between the high TTE values (red contour) and local temperature minima (blue contour) in
the cases discussed above reveals a correspondence with $HNO_3$ uptake, but a frequent lack of coincident PSC detections. The



local temperatures are just as low (and sufficient for STS formation) as in the areas outside the overlaps, but the `STS` class is not seen at all, whereas the TTE increases substantially. The corresponding detailed time histories of the temperatures at 32 hPa are shown in Figure 8d and reveal that the remarkable asymmetries in the along-track location of $HNO_3$ uptake with respect to the local temperature minimum distribution can be understood in terms of the different rates of cooling of particular

air parcels. This indicates that in the regions where slow cooling forms NAT first (marked by long white arrows in cases 2 and 4), $HNO_3$ is sequestered into NAT, and therefore STS cannot grow. In contrast, there are regions where fast cooling occurs and STS forms without prior NAT formation (marked by short white arrows in cases 2 and 4). The NAT exists as large particle / low number density clouds (sedimenting, sub-visible `MIX1`) that contain enough condensed $HNO_3$ to be detectable through MLS gas-phase depletion, but have low lidar backscatter and are invisible to CALIOP. Alternatively, it could be argued that

the NAT particles grew so large further upstream that they were effectively removed to lower altitudes through sedimentation at the time of the orbit crossing observations, leaving behind a permanently denitrified air mass detected by MLS but without coincident CALIOP PSC detections. However, further averaging of CALIOP backscatter (as discussed in Section 3) on 27 May (not shown) does indicate a considerably larger area of `MIX1` class, and so it appears that we are dealing with the limit of the L2PSCMask detection range.

## 4.2   Denitrification and renitrification

The Antarctic gas-phase $HNO_3$ distribution shown in Figure 9 provides a record of the effects of the formation and dissipation of PSCs and is displayed as the daily areal coverage for equivalent latitudes less than $60°$ S, for isentropic levels from 340 to 500 K. The white solid lines indicate low, median and high values of the $HNO_3$ probability density function. The minimum $HNO_3$ mixing ratios (i.e. the colored region below the 10th percentile white line) indicate that PSCs can lead to a complete

removal of the available ambient $HNO_3$ from the gas phase. The temperature decrease starts from the upper levels of the vortex and descends over time, resulting in PSCs and $HNO_3$ uptake developing later at the lower levels. The maximum $HNO_3$ mixing ratios (i.e. the colored region above the 90th percentile white-line) indicate episodes of renitrification at the lower levels arising from the sedimentation of PSCs. As the NAT particles fall through lower levels, they may pass into regions where they are less thermodynamically stable. In these cases the evaporative release of $HNO_3$ from the condensed phase

increases the gas-phase values, and this is detected as a rise in the $HNO_3$ measured by MLS at the lower levels. The process is seen quite clearly in the increasing time lag between the appearance of anomalously large $HNO_3$ values above the 90th percentile at 420 K (mid-May) compared to 340 hPa (mid-June). This process acts to raise the NAT temperature existence threshold at the lower levels because of the enhanced gas-phase $HNO_3$ and therefore increases the likelihood of occurrence of NAT. Heterogeneous chemical reactions on the PSC particles involving $ClONO_2$ and $N_2O_5$ produce additional $HNO_3$, which

remains in the condensed phase until the PSC dissipates (Turco et al., 1989). Therefore, an increase in gas-phase $HNO_3$ at a given level may arise from the evaporation either of sedimenting PSCs from above, or from extant PSCs that release the excess $HNO_3$ formed as a product of heterogeneous reactions. As the Antarctic winter progresses and the temperatures at the lower levels decrease, the redistributed $HNO_3$ is itself subject to further uptake into STS/NAT, resulting in potential further denitrification and consequent removal of $HNO_3$ from a larger vertical range of the lower stratosphere.





## 5 Timeseries of PSC formation and Lagrangian temperature history

In this section we examine the formation of PSCs and the Lagrangian temperature history during the early 2009 Antarctic PSC season. Figure 10 shows a timeseries of data taken over the 20 days from 2009d132 to 2009d151 (12-31 May) of $HNO_3$, TTE (temperature history), `LS_index`, and PSC fraction. Averaging bins are as described in Section 4.1. There are four columns in

each panel corresponding to four pressure levels at 68, 46, 32, and 21 hPa. Each "mini-plot" (e.g. identified by $HNO_3$ 2009d132 68hPa) is a raster image with the axes shown in Figure 2 (the reduction in plot size merges together the square pixels; there is no longer any visible white space between the measurement locations). The first column (68 hPa) for $HNO_3$ consists of 20 such images, one for each day stacked one above the other. The second column is plotted adjacent and corresponds to $HNO_3$ at 46 hPa and so on for the third (32 hPa) and fourth (21 hPa) columns. TTE is the Lagrangian temperature threshold exposure as

discussed in connection with Figure 8. The `LS_index` is as described before for each averaging bin (gray shading indicates no data, e.g. instrument off or daytime CALIOP observations; olive green shading indicates operations but no detections). PSC fraction is the ratio of the number of PSC detections to the total number of observations in each averaging bin.

Significant TTE first appears on the 32 hPa level close to the South Pole on 2009d134/2009d135 (14-15 May) and gradually increases in area and vertical extent to the 46 and 21 hPa levels over the next several days. Note the lack of corresponding areal

coverage of PSCs, especially on the 32 hPa level (only a few PSC detections are scattered about, see PSC fraction panel). The PSC `LS_index` indicates some predominantly solid class PSC detections at 68 and 46 hPa before 2009d141 (21 May), but there are fewer detections at 32 hPa. However, MLS indicates $HNO_3$ uptake on the 32 hPa level starting earlier from 2009d135 (15 May) that is as widespread as that on the 46 hPa level and similar in areal extent to the TTE. At 68 hPa there is evidence for an increase in $HNO_3$ following 2009d139 (19 May), presumably due to renitrification from evaporating NAT PSCs that have

sedimented from a higher level. The combined data at 32 hPa indicate that, in the early period, the CALIOP L2PSCMask is not detecting PSCs, since there is a persistent area of daily $HNO_3$ depletion that is not matched by corresponding PSC detections, but is consistent with the temperature history. We use the term sub-visible PSCs to refer to these cases (i.e. significant depletion of gas-phase $HNO_3$, but without detection by CALIOP).

We examine the variation of MLS $HNO_3$ with TTE for sub-visible PSCs in the period 2009d135-2009d144 (15-24 May) in

more detail in Figure 11. The scatter plot of the gas-phase $HNO_3$ values shows a range of $\sim$4 ppbv (from $\sim$14 to $\sim$18 ppbv) for low TTE, and as TTE increases beyond $\sim$1 day, the $HNO_3$ decreases. A highly simplified microphysical model has been used to calculate the uptake of $HNO_3$ from the gas phase. Growth of NAT by vapor deposition has been calculated in the manner outlined by Toon et al. (1989) and following the simplifications introduced by Carslaw et al. (2002). Here, we ignore sedimentation and calculate the growth of monodisperse NAT in a constant temperature atmosphere ($T_{ICE} + 4$ K) for two

different initial NAT densities ($5 \times 10^{-4}$ and $5 \times 10^{-5}$ cm$^{-3}$) and two initial total $HNO_3$ values that encompass the observed range (14 and 18 ppbv). The NAT growth model is initialized with a 0.1 µm radius at the nucleation time (TTE=0), and the subsequent time evolution of the gas-phase $HNO_3$ is plotted as colored curves indicating the NAT particle radius. These two curves practically bound the MLS observations of the distribution of gas-phase $HNO_3$ as a function of TTE. As is well known, low NAT number densities produce large NAT particles (Jensen et al., 2002), which take several days to reach thermodynamic





equilibrium (the uptake curves bottom out after 15 days or more). Backscatter calculations for NAT with characteristics of the upper curve (assuming $\epsilon = 0.9$) show that the backscatter (perpendicular or total) is below the CALIOP threshold of detection along the entire time evolution. For the lower curve, the higher NAT number density limits the particle growth to a much smaller radius, and backscatter calculations suggest that the NAT particles should be detectable after 0.8 days following nucleation for

the perpendicular backscatter threshold and after 1.3 days for the total backscatter threshold. The details of the scattering calculations depend on assumptions about the underlying particle characteristics, for which we lack definitive knowledge. However, observationally we repeatedly see large depletions of $HNO_3$ without accompanying particle detections, which can be accounted for qualitatively by a NAT population characterized by low number densities/large radii.

## 6  Interannual variations in the early Antarctic PSC season

In this section we investigate the interannual variability of Antarctic PSCs and $HNO_3$ over the past decade and again make use of the geolocated raster plot format discussed in Section 2.4. Figure 12 shows the MLS $HNO_3$ and CALIOP `LS_index` at 32 hPa for 30 days from day number 132 to 161 (12 May to 10 June) for the years 2006-2015. The corresponding time series for TTE and $T - T_{ICE}$ are shown in Figure 13. Additionally, in Figure 14, we present a complementary side-by-side comparison of the same observations by plotting the MLS $HNO_3$ as a time-series with each observation colored acccording to

the TTE (Figure 14a) or the CALIOP `LS_index` (Figure 14b). The evolution of the Antarctic data set of $HNO_3$, `LS_index` and the corresponding TTE and $T - T_{ICE}$ can then be followed as a function of time by reference to the above figures. In this analysis we have not treated ice PSCs separately and they are counted as part of the solid PSC population. However, in general, the number of PSCs classified as ice types in the time period considered here is low and constitutes less than 2% of the total PSC detections, except for the years 2007 and 2011 in which ice is 6% of the total and accompanying dehydration is observed

about a week before the end of the time period.

    The start of the PSC season appears to display two modes (see Figures 12b and 14b), with some years having larger coverage of solid PSCs (2007, 2011 and 2014) than others, which show larger coverage of liquid PSCs (2008, 2009, 2010 and 2015). We identify the presence of sub-visible PSCs by a delay between the onset of gas-phase $HNO_3$ depletion (see Figures 12a and 14a) and the geophysically associated detection of PSCs (see Figures 12b and 14b). Table 1 lists the day number of the initial onset

of $HNO_3$ depletion, the first day of detection of PSCs by CALIOP, the presence of sub-visible PSCs, and the predominant PSC class (solid or liquid) for the first few days following detection. Sub-visible clouds are present in five out of the seven years with MLS and CALIOP overlap during the critical time period when uptake of gas-phase $HNO_3$ is first detected. The time lag between the observed depletion of gas-phase $HNO_3$ and the detection of PSCs varies and is largest in 2009 and 2014. We infer that the sub-visible PSCs are composed of NAT because the expected uptake of $HNO_3$ is negligible for STS

at the associated ambient temperatures (see Figure 13b). In 2015 there is a notable period of several days (148–153) without many PSC detections (see Figures 12b and 14b), although $HNO_3$ is still consistently low (see Figure 12a), probably indicating sedimentation of NAT and permanent denitrification. However, since Woiwode et al. (2014) have reported the presence of NAT





with larger apparent sizes compared to compact spherical particles and with concomitant reduced settling rates, we must also consider the possibility that the NAT radius grew larger and subsequently evolved into sub-visible NAT.

The goal of describing the interannual variability of PSC seasons within the context of a decadal climatology can be met by synthesizing the CALIOP and MLS observations, consisting of the 4-variable data set (HNO$_3$, TTE, $T-T_{ICE}$ and LS_index),

into a number of characteristic groups or clusters and thereby capturing the major characteristics of PSC formation. Simple partitioning obtained by setting a single threshold boundary on each of the four variables would result in 16 different PSC groups (some of which may be empty). Such a large number of groupings would complicate interpretation, and so a more parsimonious scheme is desirable to enforce a reduction in the number of groups. Cluster analysis (CA) is a well-established tool for unsupervised exploration of a data set (Jain, 2010). Ideally members of the same cluster (identified by their proximity

to the cluster centroid) will naturally exhibit similar intrinsic properties and display low within-cluster variance, whereas the differences amongst the members of one cluster to another will display high between-cluster variance. We used a K-means centroid algorithm with a Euclidian distance metric to partition the 4-variable data set after performing zero mean and unit variance normalization. K-means analyses were performed for a varying number of imposed clusters (K=2 to K=20), and the gap-statistic (Tibshirani et al., 2001) was used to assess the optimal number of clusters by comparison with uniformly random

data and an uncorrelated data set obtained by randomly reordering the measurement times of the four variables. A steep rise in the gap-statistic was observed over the range K=2 to K=7, followed by a flattened plateau region out to K=20 (not shown). The clusterings obtained for K=6 are shown in Table 2 and Figure 15 for all the years 2006-2015. The K=6 case was chosen because the clusters fell practically into two sets (a,b) of three groups, corresponding to predominantly liquid (Figure 15a, LS_index $> 0$) and predominantly solid (Figure 15b, LS_index $< 0$) PSCs. No particular improvement was observed for the K=7 case.

The LS_index vs day number scatter plot indicates that there is some cross-over of liquid/solid PSCs present in both the a1 and a2 groups. The groups are presented superposed as a scatter plot on the thermodynamic HNO$_3$ vs $T-T_{ICE}$ diagram and show clearly that the two sets (a,b) are associated with the STS and NAT equilibrium branches. As in previous analyses, the NAT population exhibits considerable non-equilibrium effects (L12, P13), with the temperature distribution extending to temperatures as low as the STS branch. No correction has been made for the suspected temperature bias in the meteorological

temperature data (Section 3.2).

The stereographic map projections in Figure 15 show the PSC groups with higher temperatures (a3, b2, b3) extending to lower latitudes, reflecting the general outward radial increase of the polar temperature distribution. However, in particular, group b3 displays similarities to a circumpolar NAT belt (Tabazadeh et al., 2001; Höpfner et al., 2006)), although it is notable that this group also includes CALIOP detections at temperatures apparently higher than the NAT existence threshold $T_{NAT} =$

$T_{ICE} + 7\ K$. The b3 group temperature history (HNO$_3$ vs TTE) also shows little exposure to low temperatures. These may result from the inability of the gridded synoptic reanalysis data to capture accurately the local temperature minima. Another possibility is that the NAT particles may survive for a time before fully melting as they are advected downstream (L12) away from their gravity-wave sites of origin.

Compelling evidence of the differences in the formation of liquid and solid PSCs can be deduced from comparisons of the

temperature history of the groups. For the case of moderate gas-phase HNO$_3$ depletion, the occurrence of liquid PSCs (a3)



shows a steep fall off in the TTE distribution beyond ∼1 day. In contrast, the occurrence of solid PSCs (b2) extends to TTE ∼ several days and also shows a trend of declining $HNO_3$ with increasing TTE that can be explained by reference to the NAT growth curves modeled in Figure 11. For the case of large gas-phase $HNO_3$ depletion, the b1 group shows a peak around TTE=6 days, and very few of these solid PSCs formed at TTE < 2 days. This is also a consequence of the microphysics of

NAT growth since it requires ∼days for the NAT particles to grow sufficiently to substantially deplete the gas-phase $HNO_3$. Permanent denitrification caused by sedimentation of large NAT is the likely cause of the fall off in the occurrence of PSCs in the b1 group at large TTE. In contrast, there exists a substantial occurrence of liquid PSCs at all values of TTE in groups a1 and a2, as expected since the growth/evaporation of STS is a faster process dependent on the instantaneous temperature.

The total number of coincident MLS and CALIOP observations over the 10-year time span is $\sim 2 \times 10^4$, and the mean

(standard deviation) of the relative number of observations (in percent) for group a is 55 (8) and for group b is 45 (11). Therefore liquid PSCs (group a) are the dominant form and solid PSCs (group b) have slightly more variability. The corresponding breakdown of the mean and standard deviation for the individual groups is given in Table 2.

The observations in each cluster obtained from the decadal climatology were sub-divided to produce data for the individual PSC seasons. The results are shown in the pie charts in Figure 14 to provide a convenient summary of the interannual variations.

Note that the year 2006 cannot be compared to the others because the sampling consists of only three days at the end of the season. The pie charts show that the years 2009 and 2010 stand out as the ones with the highest occurrence of liquid PSCs (purple-blue sectors) (77% and 83%, respectively) in the 10-year record. The remaining years fall into two broadly related categories with a slightly larger proportion of liquid PSCs in 2007 (52%), 2008 (54%), 2013 (58%) and 2015 (53%) and a slightly larger proportion of solid PSCs (red-yellow sectors) in 2011 (63%), 2012 (58%) and 2014 (57%).

## 7   Conclusions

A decade of MLS and CALIOP satellite measurements from 2006 to 2015 over the Antarctic polar vortex were used to investigate the early season development of PSCs and the gas-phase $HNO_3$ distribution in the lower stratosphere. We developed a compact visual representation of the daily orbit tracks that allows a time-series to be constructed from the montage of a few hundred separate daily images, consisting of a combination of different days/pressure levels/species, and displayed on a single

page.

Lidar properties were calculated for the STS and NAT components of PSCs. We reviewed the capabilities of spaceborne instruments (lidar, mid-infrared, microwave) applied to the detection of PSCs and uptake of $HNO_3$ and used the results to investigate the generation of large particle / low number density NAT at temperatures above the ice frost point, which results in sub-visible PSCs that can be detected through the gas-phase uptake of $HNO_3$. The presence of sub-visible PSCs was found

in over half of the years examined.

In this work we have demonstrated that the early initiation of NAT nucleation in the Antarctic vortex takes place frequently at temperatures above the ice frost point and often before cooling produces liquid PSCs. A consistent picture emerges, with $HNO_3$ depletion occurring in the inner vortex, usually before any associated PSC development is detected by CALIOP. We conclude




that an ice-seeding process is not essential for the initiation of NAT nucleation or subsequent development of large-scale NAT growth in the early winter.

We used detailed measurements in the CALIOP and MLS along-track transects to illustrate that the formation of PSCs is governed not only by the local ambient temperature, but is also shaped in large measure by the underlying temperature histories.

A cluster analysis method was used to organize the combined CALIOP and MLS data into a manageable form to guide an investigation of the decadal climatology and facilitate ready comparison of the patterns of interannual variability. The temperature distribution of the groups was used to compare the relative frequency of formation of liquid and solid PSCs in the Antarctic lower polar stratosphere.

*Acknowledgements.* We gratefully acknowledge members of the teams associated with the CALIOP and MLS instruments, and the GEOS-
5 meteorological analyses. MLS data are archived at the NASA Goddard Earth Sciences Data Information and Services Center. CALIOP data were obtained from the NASA Langley Research Center Atmospheric Science Data Center. IDL software for calculation of PSC thermodynamic properties provided by M. E. Hervig was obtained from the GATS Scientific Software website (http://gwest.gats-inc.com/software/software_page.html). Fortran software for $T$-matrix calculations provided by M. I. Mishchenko was obtained from the NASA GISS website (http://www.giss.nasa.gov/staff/mmishchenko/t_matrix.html). Work at the Jet Propulsion Laboratory, California Institute of
Technology, was carried out under a contract with the National Aeronautics and Space Administration.



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





**Table 1.** Comparison of times of occurrence of $HNO_3$ depletion and detection of PSCs at the start of each PSC season.

| Year | Onset day of $HNO_3$ depletion | First day of PSC detection | Sub-visible PSCs | Dominant PSC Class |
|------|------|------|------|------|
| 2006 | 148 | † | † | † |
| 2007 | 137 | 137 | No | SOL |
| 2008 | 145 | 147 | Yes | LIQ |
| 2009 | 135 | 141 | Yes | LIQ |
| 2010 | 140 | 141 | Yes | LIQ |
| 2011 | 140 | 141 | Yes | SOL |
| 2012 | 139 | † | † | † |
| 2013 | 138 | † | † | † |
| 2014 | 133 | 139 | Yes | SOL |
| 2015 | 139 | 139 | No | LIQ |

† indicates no MLS and CALIOP overlap





**Table 2.** Results of the K=6 cluster analysis for 2006–2015: the number of members assigned to each group, the mean and standard deviation of the relative number of observations and the centroids in the 4-variable data space are given.

| Group Name | LIQ/SOL | Number | Mean (%) | Standard Deviation (%) | $HNO_3$ (ppbv) | TTE (days) | $T - T_{ICE}$ (K) | LS_index |
|---|---|---|---|---|---|---|---|---|
| a1 | LIQ | 3194 | 14 | 8 | 1.41 | 11.22 | 0.07 | 0.56 |
| a2 | LIQ | 5139 | 23 | 7 | 2.84 | 3.30 | 0.67 | 0.55 |
| a3 | LIQ | 3532 | 18 | 9 | 11.19 | 0.88 | 3.10 | 0.60 |
| b1 | SOL | 3194 | 16 | 10 | 1.33 | 6.80 | 0.71 | -0.73 |
| b2 | SOL | 4206 | 24 | 10 | 9.04 | 1.61 | 4.00 | -0.83 |
| b3 | SOL | 866 | 5 | 1 | 13.25 | 0.02 | 10.97 | -0.90 |





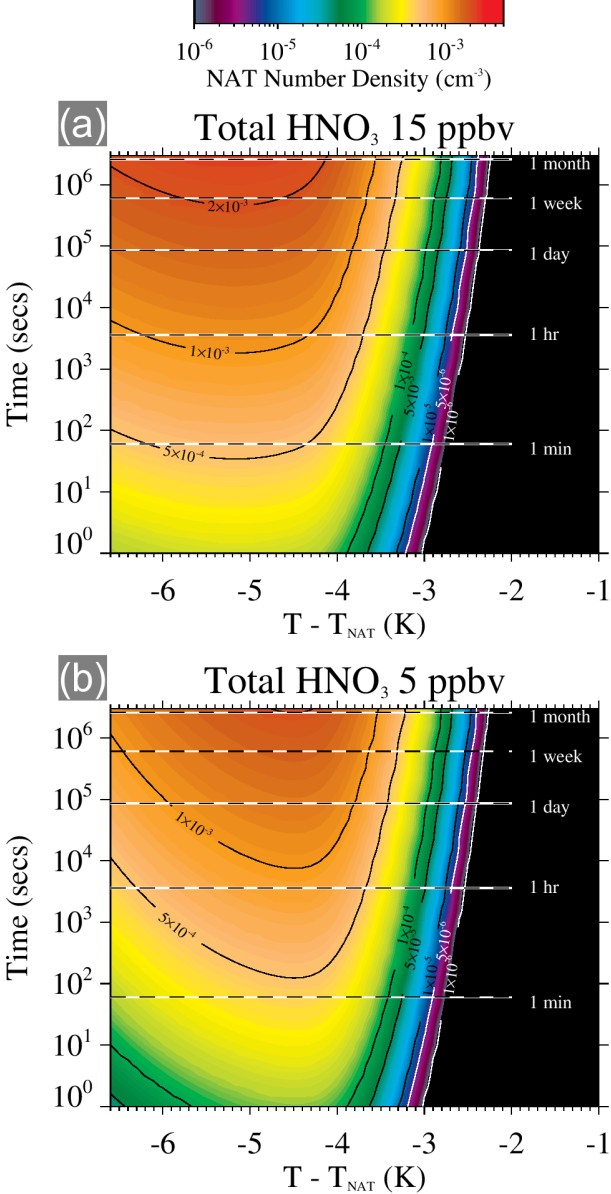

**Figure 1.** (a): Calculated NAT number densities (colored shading and labeled contour lines) resulting from varying temperature exposure durations in an unperturbed atmosphere with 15 ppbv total $HNO_3$. Horizontal dashed lines highlight exposures of between 1 minute and 1 month. An air parcel (containing the requisite background aerosol embedded with foreign nuclei) exposed to a temperature of $T - T_{NAT} = -4$ K for one day will generate 0.0012 cm$^{-3}$ of nucleated NAT particles. (b): as for (a) except for a denitrified atmosphere with 5 ppbv total $HNO_3$.









**Figure 2.** (a): Polar stereographic projection of the MLS measurement locations for all orbits over the Antarctic on 2009d146 (26 May 2009). Each orbit is numbered sequentially. The square symbols denote the latitude and longitude locations of the MLS vertical profiles. The size of the squares is not representative of the along-track or across-track resolutions. $HNO_3$ volume mixing ratio at 32 hPa is given by the color bar. (c): The same data points are shown as a temporal raster plot. The ordinate is the time of day in hours (UT) and the abscissa is the geodetic along-track angle. The squares denote the time of day of the MLS measurement and the measurement location with respect to the closest approach of the orbit track to the South Pole. Each orbit is numbered along the right ordinate. Also shown are the along-track distance, the latitude and the solar zenith angle. The size of the squares is not representative of the MLS integration time or the along-track resolution. (b,d): as for (a,c) except for 2009d156 (5 June 2009).



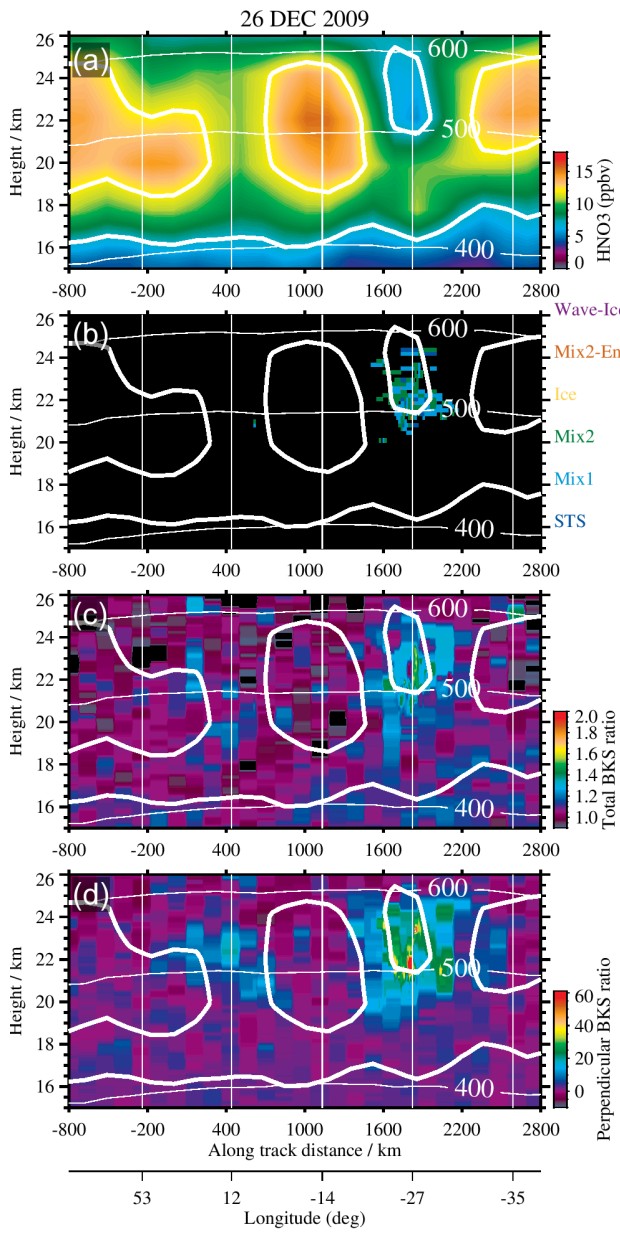

**Figure 3.** Comparison of along-track data for the partial orbit shown by Hoyle et al. (2013) (note that the x-axis is reversed here from their figure). (a) MLS $HNO_3$ showing sequestration at $12°$ and $-27°$ longitude. (b) CALIOP PSC Mask does not show detection of PSCs at $12°$ longitude. (c) Smoothed CALIOP total backscatter ratio. (d) Smoothed CALIOP perpendicular backscatter ratio. Solid thick white contours are the MLS $HNO_3$ isolines for 7 and 12 ppbv $HNO_3$. Solid thin white vertical lines are the longitude markers shown by Hoyle et al. The detection of PSCs near $12°$ longitude is evident in the smoothed CALIOP perpendicular backscatter ratio along with the corresponding $HNO_3$ sequestration measured by MLS in panel (a).





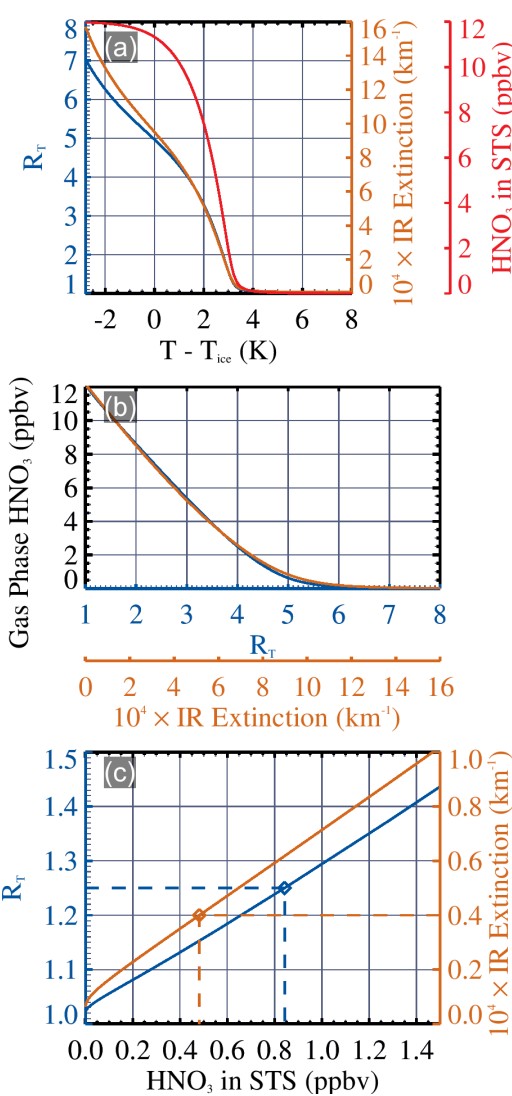

**Figure 4.** (a) Temperature variation relative to the frost-point of the uptake of $HNO_3$ in STS (red), the calculated 532 nm lidar total backscatter ratio, $R_T$, (blue), and the 12 μm infrared extinction (orange). (b) Gas-phase $HNO_3$ vs 532 nm total backscatter ratio (blue) and 12 μm infrared extinction (orange). (c) STS detection limits for lidar backscatter ratio (blue diamond) and infrared limb extinction (orange diamond) and correspondence to the uptake of $HNO_3$ in STS.



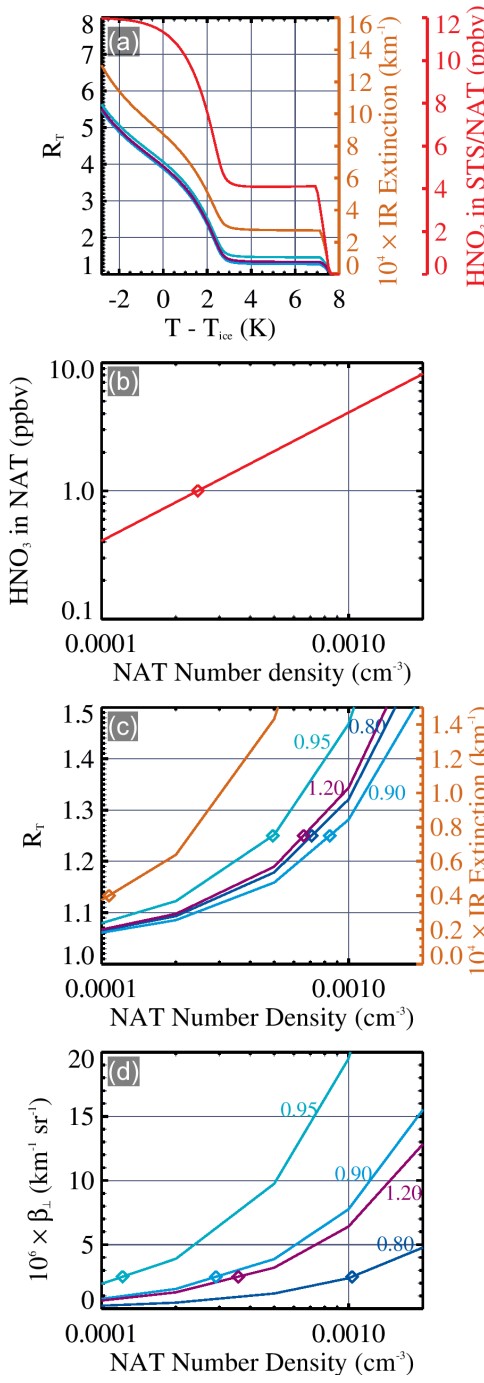





**Figure 5.** (a) Temperature variation relative to the frost-point of the uptake of $HNO_3$ in an STS/NAT mixture (red) for a NAT number density of 0.001 cm$^{-3}$ and an effective radius of 6.5 µm, the calculated lidar backscatter ratio for four different particle shape aspect ratios (purple-blue), and the 12 µm infrared extinction (orange). (b) Condensed $HNO_3$ in NAT (red line) at $T - T_{ICE} = 5$ K as a function of NAT number density. (c) NAT detection limits for lidar total backscatter ratio (blue-purple diamonds) and infrared limb extinction (orange diamond) and correspondence to the NAT number density. (d) as (c) except for the lidar perpendicular backscatter coefficient only.





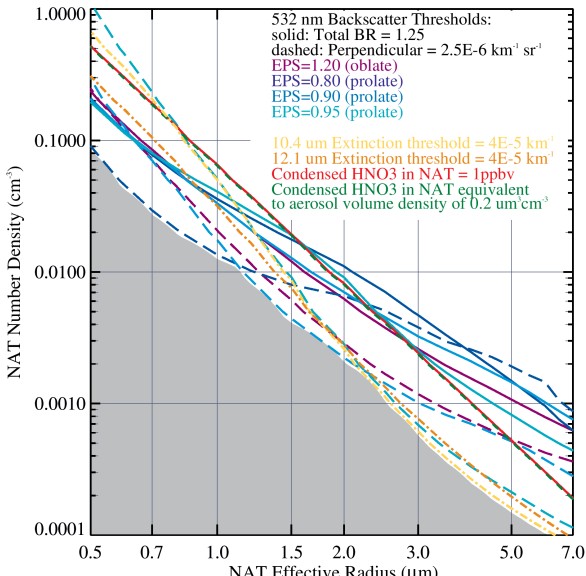

**Figure 6.** Intercomparison of the sensitivity of various PSC detection techniques to a range of NAT number densities and effective radii at a temperature of $T - T_{ICE} = 5$ K, for an ambient pressure of 46 hPa, 5 ppmv $H_2O$, and 12 ppbv total $HNO_3$. The purple-blue lines indicate the limits of detection for lidar with total backscatter (solid) and orthogonal channel (dashed) for a range of aspect ratios (EPS). The red (green) line indicates the sensitivity of an in situ sampling instrument to 1 ppb of condensed $HNO_3$ in NAT (aerosol density, 0.2 μm³ cm⁻³) or the equivalent uptake from the gas phase (e.g. by a microwave limb sounder). Yellow/orange lines indicate the sensitivities in the mid-infrared for a limb sounder at two wavelengths. NAT PSCs with particle distribution characteristics lying within the gray shaded region are undetectable by any of the above techniques, with the assumed horizontal and vertical averaging scales given in the text.









**Figure 7.** CALIOP PSC type classifications and their corresponding 2-D cross-section pairs derived from the 3-D coordinate space of temperature $T - T_{\mathrm{ICE}}$, $HNO_3$ and total backscatter, $R_T$, for 10 May to 25 October 2009. The six columns are the CALIOP PSC types identified in the text. Blue (green) lines are theoretical calculations for total $HNO_3$ from 2 to 24 ppbv in 2 ppbv steps for STS (NAT) equilibrium. Red-black dashed lines are theoretical calculations for equilibrium NAT with number densities 0.001 cm$^{-3}$ (bottom curve) and 0.01 cm$^{-3}$ (top curve) and 14 ppbv total $HNO_3$. The lower limit of detection, given by the black-white dashed line, is described in the text.









**Figure 8.** Colocated MLS and CALIOP orbit transects for selected orbits showing in six rows: (a): MLS $HNO_3$. (b): ambient temperature, $T-T_{ICE}$. (c): Temperature threshold exposure, TTE (days). (d) 15-day reverse trajectory temperature history, $T-T_{ICE}$ ending at the 31 hPa pressure level. Minimum temperature encountered along the trajectory is inset at the top left. (e): CALIOP L2PSCMask PSC Classification. (f): LS_index. MLS pressure levels are shown as labeled black or white contours. Blue (green) contours indicate $T_{ICE} + 2$ K ($T_{NAT}$). Red contours indicate TTE values $\geq 3$ days. The MLS 12 ppbv $HNO_3$ contours are indicated in yellow.





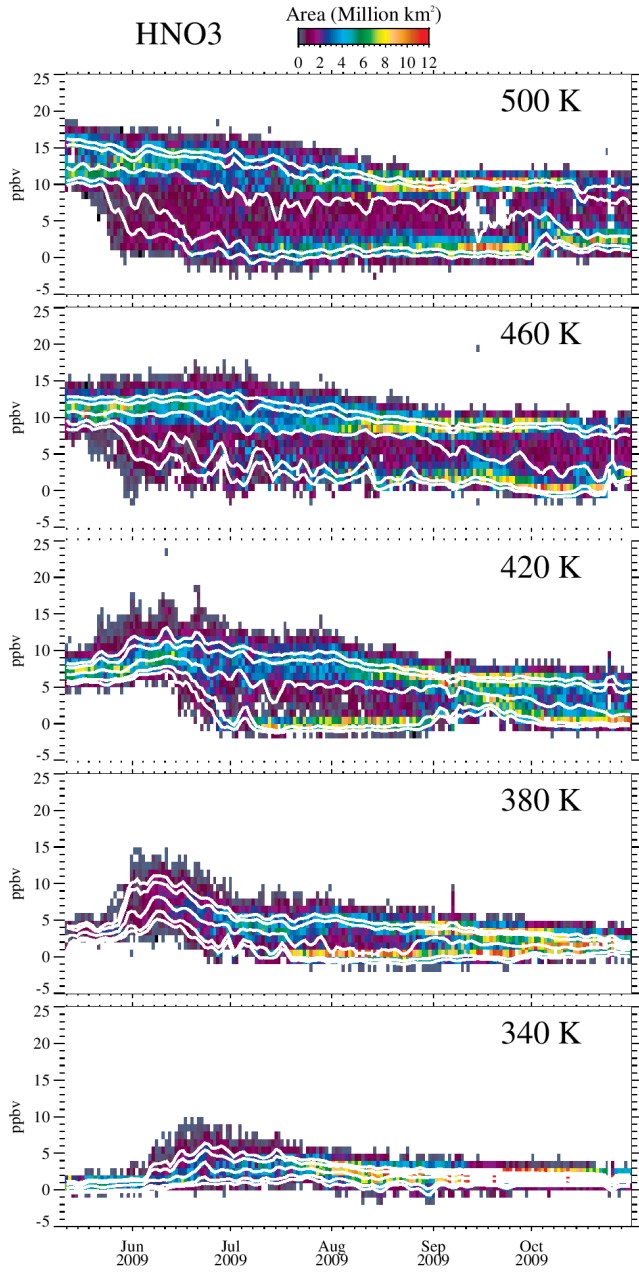

**Figure 9.** Time-series of the distribution of MLS $HNO_3$ in the Antarctic from May to October 2009, for equivalent latitudes less than 60° S, and for isentropic levels from 340 to 500 K. The color scale indicates the areal coverage. White solid lines indicate the 10th, 20th, 50th (i.e. median), 80th, and 90th percentiles of the $HNO_3$ probability density function.





**Figure 10.** 20-day time-series of Antarctic raster plots at 68, 46, 32 and 21 hPa in 2009. Gray shading indicates no observations, olive-green shading indicates observations but no detections. (a): MLS $HNO_3$. Numbers on the right axis indicate the median $HNO_3$ for the region where TTE > 2 days (in ppbv) for each day at 32 hPa. (b): Temperature history. A set of three numbers on the right axis for each day at 32 hPa indicates in ascending order (i) maximum TTE in days, (ii) minimum local temperature, $T - T_{ICE}$ in K, (iii) minimum temperature encountered along the back-trajectory, $T - T_{ICE}$ in K. (c): CALIOP `LS_index`. (d): CALIOP PSC fraction. Numbers on the right axis indicate the maximum PSC fraction (ratio of number of PSC detections to number of observations) for each day at 32 hPa.





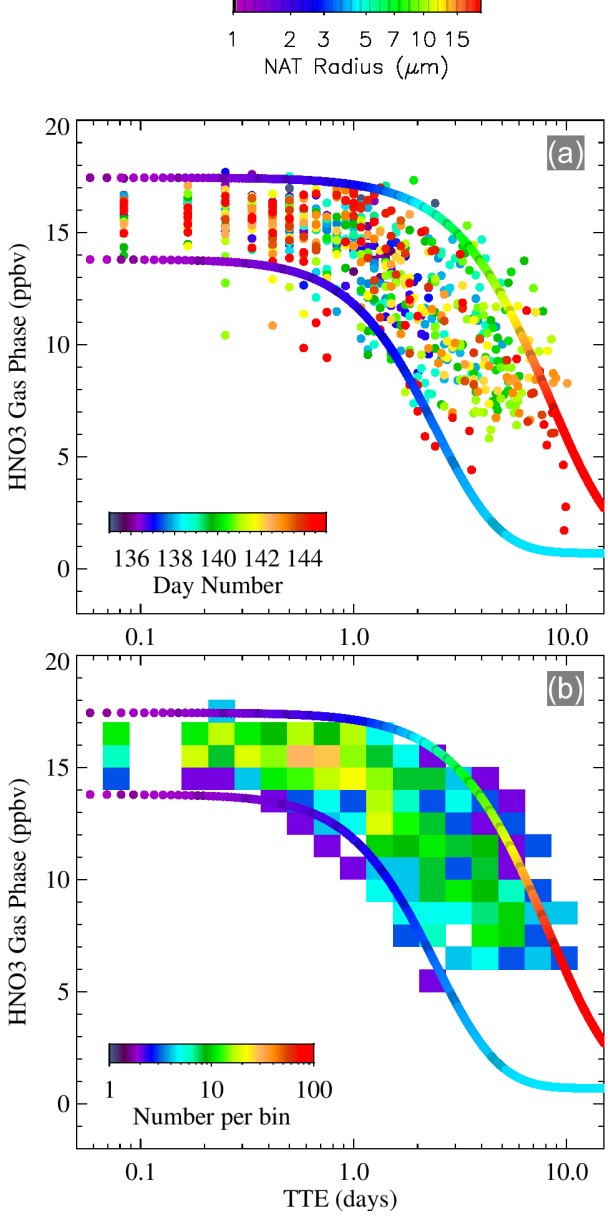

**Figure 11.** MLS $HNO_3$ on the 32 hPa pressure level over the 10-day period from 15-24 May 2009 (days 135-144) in the Antarctic polar vortex. Data are selected only if there are no coincident PSC detections by CALIOP. (a): Scatter plot of individual $HNO_3$ values vs TTE. The colored dots indicate the measurement day number (given in the inset color bar). The two colored curves are from a calculation of the gas-phase $HNO_3$, assuming growth of NAT at a constant temperature of $T_{ICE} + 4$ K for NAT number densities and initial total $HNO_3$ values of $5 \times 10^{-5}$ cm$^{-3}$ and 18 ppbv, respectively, for the upper curve and $5 \times 10^{-4}$ cm$^{-3}$ and 14 ppbv for the lower curve. The top color bar indicates the NAT radius. (b): As in (a) except plotted as a 2-D histogram of the density of points, with the inset color bar giving the number of samples per bin. Only bins accumulating two or more data points are shown.



**Figure 12.** 30-day time-series of Antarctic raster plots at 32 hPa for 2006–2015. Diamond symbols indicate the last day before the start of detectable HNO$_3$ depletion. Gray shading indicates no observations, olive-green shading indicates observations but no detections. Day number is shown on the vertical axis. (a): MLS gas-phase HNO$_3$. (b): CALIOP `LS_index`.





**Figure 13.** 30-day time-series of Antarctic raster plots at 32 hPa for 2006–2015. Day number is shown on the vertical axis. (a): TTE (days).

(b): $T - T_{ICE}$ (K).









**Figure 14.** Time-series for the Antarctic for associated $HNO_3$ and PSCs during 2006–2015 at 32 hPa. (a): MLS $HNO_3$ vs day number. The color scale shows the TTE. Only data for TTE > 0.1 day are plotted. The inset pie charts indicate the relative proportions of measurements in six categories (red-yellow sectors are solid PSCs, purple-blue sectors are liquid PSCs) as defined in the text and in Figure 15. (b): CALIOP `LS_index` vs day number. The color scale shows the `LS_index`. Gray shading indicates no measurements. Light green shading in (b) indicates the envelope of MLS $HNO_3$ observations in (a).









**Figure 15.** Results of the K=6 cluster analysis on the combined MLS and CALIOP data for the Antarctic for the years 2006-2015 at 32 hPa.
Thermodynamic diagrams indicate for reference the theoretical $HNO_3$ uptake by STS (blue-black dashed line) and by NAT (green-black
dashed line). The scatter plots show the MLS $HNO_3$ vs temperature relative to the ice frost point colored by the corresponding `LS_index`
obtained from CALIOP. Column a (b) shows the data in three groups a1-a3, (b1-b3), which are seen to be predominantly liquid (solid)
PSC detections according to the `LS_index`. The colored contours ((a1-purple, a2-dark-blue, a3-light-blue), (b1-red, b2-orange, b3-yellow))
contain 95% of the corresponding cluster members. The maps show the geographic distribution of the members of each of the clusters and
are color-coded by the `LS_index` (inset color bar). A 2-D view ($HNO_3$ vs TTE) is provided of the cluster densities (purple-yellow shading
indicates low to high density) and the accompanying colored solid lines represent the normalized 1-D histogram of the TTE values. The
cluster densities are also displayed as a 2-D view of `LS_index` vs day number.