# Peer review of "Interannual variations of early winter Antarctic polar stratospheric cloud formation and nitric acid observed by CALIOP and MLS"

_Atmospheric Chemistry and Physics, 2016_

## Referee Comment (RC1) · Anonymous Referee #1 · 11 Jul 2016

**General comments**

In this paper, Lambert et al. investigate in detail the start of the PSC season in the Antarctic stratosphere with the aim to explain the early formation of nitric acid trihydrate (NAT). For this, a decade of quasi-simultaneous observations of HNO3 (and H2O) by the MLS instrument on Aura and PSC measurements by the CALIPSO lidar are utilised. The authors argue that observed early loss of HNO3 from the gas-phase without co-incident detection of PSCs by CALIPSO is due to the formation of large NAT particles of low number density. These findings are corroborated by trajectory calculations indicating temperatures some Kelvin below the NAT equilibrium temperature as a prerequisite for the occurrence of such situations. This supports early NAT formation mechanisms

without the necessity for the existence of ice particles, thus backing previous observations in the Arctic wintertime stratosphere. Further, a comprehensive overview of ten years of MLS and CALIPSO early-Antarctic wintertime observations shows the relative strength of NAT versus STS formation in each year.

The paper is well written and conclusive. Especially I would like to point out the effort the authors have taken in the development of new methods of data presentation to support their argumentation. Thus, I strongly support its publication in ACP.

The only major points I would like the authors to consider are: (1) is the discussion of the IR limb-instruments' sensitivity of any help for the flow of the arguments or does it perhaps distract the reader from the important points? (2) I miss a more in-depth discussion on measurement restrictions, like the accuracy of MLS and the lack of observations in the central part of the vortex, on the conclusions drawn.

**Specific comments**

P4L30: 'scaled potential vorticity'

Please add a reference for sPV or a short explanation.

P5L20: 'with single-profile precisions of 4–15 et al., 2007) and 0.7 ppbv for HNO3 (Santee et al., 2007)'

Could you also give estimates on the accuracy of H2O and HNO3 from MLS in the wintertime high-latitude regions (systematic error estimation and/or comparison with independent observations)? How strong would errors in those gases affect your following argumentation?

P6L3: 'Typical lower stratospheric polar values for TICE and TNAT are 188 K and 195 K, respectively.'

These values depend on pressure, H2O and HNO3. Could you indicate the ranges?

P8L16: 'For a mid-infrared limb sounder operating in the window-region near 12 $\mu$m

we use an extinction threshold, kext = 4×10−5 km−1 based on measurements by the Improved Stratosphere and Mesosphere Sounder (Lambert et al., 1996).' Also e.g. P8L22,P10L29

Generally, I do not understand why IR limb sounders are included in the discussion while such measurements are not used at all in the scientific argumentation of the paper. Furthermore, such a threshold depends very strongly on instrumental details like e.g. spectral noise, field-of-view and radiometric calibration accuracy. I would not assume it justified to use here thresholds determined for an instrument (ISAMS) which has not been operated in parallel to CALIPSO and MLS but to skip MIPAS and HIRDLS. In addition, after the paper by Lambert et al, 1996, there have been findings regarding simulation of limb radiances and the importance of scattered mid-IR radiation (e.g. Höpfner, 2004) which do also impact the argumentation on sensitivity (by increasing the sensitivity for particles larger than 1 $\mu$m). In conclusion, I would suggest to either go into more detail regarding the IR limb-sounding thresholds (instrumental parameters, refractive indices used for STS and NAT, forward model including scattering...), or, perhaps better, skip this discussion and concentrate on LIDAR and MLS.

P9L2: 'The morphology of NAT particles is still an open question, as is the compactness of the particles (Molleker et al., 2014; Woiwode et al., 2014).'

One could also mention here a recent publication by Woiwode et al., 2016.

P15L13: 'However, further averaging of CALIOP backscatter (as discussed in Section 3) on 27 May (not shown) does indicate a considerably larger area of MIX1 class, and so it appears that we are dealing with the limit of the L2PSCMask detection range.'

Where does this extended area lie? Below (indicating sedimentation) or in the TTE region? Maybe a figure could be added as supplemental material

P15L20 and Fig. 9: 'can lead to a complete removal of the available ambient HNO3 from the gas phase.'

Could you comment on the large amount of negative HNO3 values. Are those compatible with the MLS precision or does it hint to some systematic error?

P17, chapter 6.

I would like to see some discussion on the centeredness of the polar vortex around the S-pole between the different years and whether this correlates somehow with the presented data – especially related also to the central part, which is missed by the observations.

P40, Fig. 10:

Could you show here also T-Tice (e.g. in the appendix)? Does this look significantly different from TTE?

**Technical corrections**

P3L10: 'predictibility' -> 'predictability'

P4L8: 'superceded' -> 'superseded'

P5L17: 'aproximately' -> 'approximately'

P11L16: 'correponding' -> 'corresponding'

P34, caption Fig. 6: 'amibient' -> 'ambient'

P39, x-axis labels: Please indicate if the major tick mark labels are the beginning or the middle of the month.

P40, Fig. 10: Please add information on the real date (in addition to 2009d132, . . .)- at least in the caption for the first and last day.

**References**

Höpfner, M., Study on the impact of polar stratospheric clouds on high resolution mid-IR limb emission spectra, J. Quant. Spectrosc. Radiat. Transfer, Vol. 83, No. 1,

93-107, doi: 10.1016/S0022-4073(02)00299-6, 2004.

Woiwode W., M. Höpfner, L. Bi, M. C. Pitts, L. R. Poole, H. Oelhaf, S. Molleker, S. Borrmann, M. Klingebiel, G. Belyaev, A. Ebersoldt, S. Griessbach, J.-U. Grooß, T. Gulde, M. Krämer, G. Maucher, C. Piesch, C. Rolf, C. Sartorius, R. Spang, and J. Orphal, Spectroscopic evidence for large aspherical $\beta$-NAT particles involved in denitrification in the December 2011 Arctic stratosphere Atmos. Chem. Phys. Discuss., doi:10.5194/acp-2016-146, accepted for ACP, 2016.

---

## Referee Comment (RC2) · Anonymous Referee #2 · 12 Jul 2016

This is a complicated and comprehensive over view of what may be learned by combining co-located remote measurements of gas phase HNO3 from the Aura microwave limb sounder (MLS) and the CALIOP lidar measurements of particle backscatter and depolarization. The authors combine these measurements with back trajectory calculations to address questions related to the nucleation of nitric acid trihydrate (NAT) within polar stratospheric clouds.

The primary conclusion that there is significant NAT formation at temperatures well above Tice for NAT with low number concentrations and large particle sizes implies that particle observations confirming this conclusion are not possible due to the sensitivities of the lidar measurements. Thus a significant fraction of the paper is devoted

to microphysical calculations of the sensitivities of the various instruments to particles of varying sizes, number concentrations, and shapes. This is necessary to ultimately rely solely on the MLS measurements of HNO3 gas phase loss as evidence of PSC formation when concomitant particle observations provide no measurable PSCs. Thus the primary conclusion on NAT nucleation requires only MLS gas phase hno3 loss and an absence of PSC detection by the CALIOP lidar.

The authors present a lot of information in fairly compact forms and it is difficult to suggest any effective ways to shorten or compress the figures. The color coding can be at times difficult to fully distinguish the various categories of particle calculations and I wonder if all the aspect ratios presented are necessary. I should think the aspect ratios bounding the range are sufficient.

It is not clear why the sensitivities to IR measurements at 12 um are presented. No IR measurements are used in the analysis and none are currently available for PSC measurements to my knowledge. Is this just done for completeness? In any case the slant indicating the IR measurements are somehow more sensitive to PSCs based on the graphs shown seems a bit arbitrary. The IR emission measurements will be more sensitive to larger particles since they are sensitive to volume whereas the visible scattering measurements are sensitive to aerosol cross section so more sensitive to smaller particles.

The paper should be published with minor corrections. I provide some specific comments for consideration below.

5.6-13. This is too much detail that will make sense to the very few who know what PSC_compositions of 2 and 3 are. I would recommend a short description of what the distinction between mix1 and mix2 is and why it is important, then just explain that the separation failed to include in mix1 those observations with a scattering ratio less than 1.25. Later 1.25 is used as a threshold to identify PSC presence so overall this is a bit confusing.

6.2. "The HNO3 and H2O values (for estimating TNAT and TICE) are assumed to be the same at all back-trajectory times." This does not make sense. The same as what? Are they assumed to be constant from the start point of the back trajectory, so not updated along the trajectory, or ???

8.8. Do the authors believe there is really uncondensed h2so4 available for uptake? If so what are they using for the ambient h2so4 gas phase mixing ratio?

8.15. Rt=1.25 as a threshold? This is the same value used earlier in the revised classification scheme and now here is used as a threshold to identify PSCs. Seems a bit confusing. In addition Fig. 4 c) uses 1.25 as the threshold for STS detection, so clearly NAT PSCs could appear below this level. Please clarify.

8.29-31 and Fig. 4c). This statement and the figure don't show that the IR is more sensitive than the visible to STS content. It just shows they are different. The ordinates in Fig 4c) are somewhat arbitrarily chosen and the assumed thresholds are set on somewhat subjective criteria that are not fully justified and are different. Again for this paper what is the point of showing the IR results?

9.18-21. Is this realistic? Would the large NAT particles not continue to take up hno3, given enough time, until it was all consumed, so additional uptake of hno3 is not dependent on STS?

9.30-31. Same question as above. Why does the uptake of hno3 depend on the number density of particles? Is this some sort of kinetic limit or resident time limit, i.e. particles growing too large to stay in the air space? I am not aware there is a theoretical limit to the size a NAT particle can attain given existence at T < Tnat and hno3 and h2o available.

10.9-10. I suppose this should be no surprise since IR emission will be sensitive to volume, R^3 whereas backscatter is sensitive to cross section, R^2. Thus large particles cause larger changes in volume.

10.17 How is effective radius, Reff, defined? Is this 3V/S so the ratio of the third and second moments?

14.25. I do not see a blue line, indicating Tice+2K, on any of the panels for case 3.

15.28. I assume the authors mean 340 K (mid-June).

16.10. Where is the gray shading? I do not see any. It only appears in later figures so save the explanation for then.

16.14. It takes quite a few days for significant TTE to be observed at 21 hPa, nearly half the period shown, so the statement here comparing 21 and 46 hPa for the appearance of significant TTE is confusing.

Fig 13a). Where are the diamonds in the TTE plot?

[Figure]

---

## Author Comment (AC1) · 20 Sep 2016

**acpd-2016-433 Reply to Referee Comments**

**Early winter Antarctic PSCs and HNO$_3$ observed by CALIOP and MLS**

**A. Lambert, M. L. Santee and N. J. Livesey**

We thank the reviewers for their careful reading of the manuscript and appreciate their suggestions for improvements. We address their comments in the following text and outline our plan for submitting a revised manuscript accordingly.

Referee Comments RC1/RC2 are in blue type.

Author Responses are in red type.

**RC1 General Comments:**

In this paper, Lambert et al. investigate in detail the start of the PSC season in the Antarctic stratosphere with the aim to explain the early formation of nitric acid trihydrate (NAT). For this, a decade of quasi-simultaneous observations of HNO3 (and H2O) by the MLS instrument on Aura and PSC measurements by the CALIPSO lidar are utilised. The authors argue that observed early loss of HNO3 from the gas-phase without coincident detection of PSCs by CALIPSO is due to the formation of large NAT particles of low number density. These findings are corroborated by trajectory calculations indicating temperatures some Kelvin below the NAT equilibrium temperature as a prerequisite for the occurrence of such situations. This supports early NAT formation mechanisms without the necessity for the existence of ice particles, thus backing previous observations in the Arctic wintertime stratosphere. Further, a comprehensive overview of ten years of MLS and CALIPSO early-Antarctic wintertime observations shows the relative strength of NAT versus STS formation in each year. The paper is well written and conclusive. Especially I would like to point out the effort the authors have taken in the development of new methods of data presentation to support their argumentation. Thus, I strongly support its publication in ACP. The only major points I would like the authors to consider are: (1) is the discussion of the IR limb-instruments sensitivity of any help for the flow of the arguments or does it perhaps distract the reader from the important points? (2) I miss a more in-depth discussion on measurement restrictions, like the accuracy of MLS and the lack of observations in the central part of the vortex, on the conclusions drawn.

We will address the concerns of the referee in the modified manuscript and aim to improve the discussion of the sensitivity of IR limb instruments and the inherent measurement uncertainties in our analyses.

**RC1 Specific Comments:**

**P4L30:** 'scaled potential vorticity' Please add a reference for sPV or a short explanation.

We will include a short explanation in the text and cite an appropriate reference for sPV.

**P5L20:** 'with single-profile precisions of 4-15% Lambert et al., 2007) and 0.7 ppbv for HNO3 (Santee et al., 2007)'. Could you also give estimates on the accuracy of H2O and HNO3 from MLS in the wintertime high-latitude regions (systematic error estimation and/or comparison with independent observations)? How strong would errors in those gases affect your following argumentation?

We will address this concern by stating the estimated accuracies from MLS and providing the equivalent uncertainties in the derivation of TICE and TNAT obtained by error propagation.

**P6L3:** 'Typical lower stratospheric polar values for TICE and TNAT are 188 K and 195 K, respectively.'

These values depend on pressure, H2O and HNO3. Could you indicate the ranges?

Yes, we will provide a compact tabulation of TICE and TNAT (and their uncertainties) against pressure for typical H2O and HNO3 ranges.

**P8L16:** 'For a mid-infrared limb sounder operating in the window-region near 12 um we use an extinction threshold, kext = 4 10 5 km 1 based on measurements by the Improved Stratosphere and Mesosphere Sounder (Lambert et al., 1996).' Also e.g. P8L22,P10L29.

Generally, I do not understand why IR limb sounders are included in the discussion while such measurements are not used at all in the scientific argumentation of the paper. Furthermore, such a threshold depends very strongly on instrumental details like e.g. spectral noise, field-of-view and radiometric calibration accuracy. I would not assume it justified to use here thresholds determined for an instrument (ISAMS) which has not been operated in parallel to CALIPSO and MLS but to skip MIPAS and HIRDLS. In addition, after the paper by Lambert et al, 1996, there have been findings regarding simulation of limb radiances and the importance of scattered mid-IR radiation (e.g. Höpfner, 2004) which do also impact the argumentation on sensitivity (by increasing the sensitivity for particles larger than 1 um). In conclusion, I would suggest to either go into more detail regarding the IR limb-sounding thresholds (instrumental parameters, refractive indices used for STS and NAT, forward model including scattering: : :), or, perhaps better, skip this discussion and concentrate on LIDAR and MLS.

We would prefer to add further supporting material on the capabilities of IR limb sounders for aerosol/PSC detection. UARS ISAMS/CLAES/HALOE were of course operated in a period of enhanced stratospheric aerosol loading from Mt Pinatubo. We can also bring into the discussion the contemporary measurements from MIPAS and HIRDLS to provide a more inclusive analysis of the performance of the IR limb sounders. In particular, in a comparison of CALIOP and MIPAS PSC measurements, Hopfner et al (2009) (*Comparison between CALIPSO and MIPAS observations of polar stratospheric clouds, J. Geophys. Res., 114, D00H05, doi:10.1029/2009JD012114*) found a discrepancy in the number of Antarctic PSC detections during May.

In Lambert et al (2012) (*A-train CALIOP and MLS observations of early winter Antarctic polar stratospheric clouds and nitric acid in 2008, Atmos. Chem. Phys., 12, 2899-2931, doi:10.5194/acp-12-2899-2012, 2012*) we commented on the findings by Höpfner et al as follows . . .

> *Comparisons between CALIOP and MIPAS matching pair observations by Höpfner et al. (2009) in 2006-2007 found that overall both instruments detected Antarctic PSCs in common in 85% of the matches during June-August, but only 60% in May. In particular, there were considerably more MIPAS-only PSC detections (i.e without corresponding matching detections by CALIOP) in May (25%), compared to June (8%) and July-October (< 3%). Höpfner et al. (2009) attributed these more frequent detections to the MIPAS limb-viewing geometry, which is oriented towards the pole, and results in sensitivity to PSCs lying poleward of the tangent points, but beyond the latitude limit of the CALIOP nadir view. Since PSC formation largely occurs in a region close to the South Pole in May, the MIPAS viewing geometry has a much larger effect on the PSC detection frequency at this time compared to later times when PSCs occur over a more widespread region. Note that the MLS line of sight is in the forward direction along the orbit track and is not oriented towards the pole, therefore the low HNO3 measurements are not biased by the viewing ge-*

*ometry.*

Our analysis of Antarctic early-winter PSCs now extends to a decade of measurements and provides conclusive evidence that another factor, other than viewing geometry of cloud inhomogeneity, is also at play in the lower number of Antarctic PSC CALIOP detections in May compared to MIPAS – the common occurrence in some early winters of large particle NAT clouds which are not detected by CALIOP.

Since we submitted our manuscript a new paper on the MIPAS PSC classification scheme has been published by Spang et al (2016) (*A multi-wavelength classification method for polar stratospheric cloud types using infrared limb spectra, Atmos. Meas. Tech., 9, 3619-3639, doi:10.5194/amt-9-3619-2016, 2016.*). This paper presents an intercomparison of CALIOP vs MIPAS PSC detections and statistics on the frequency of type classifications that are relevant here and we plan to discuss these findings in our modified text. Particularly, we note that only the climatological averages of PSC count statistics over the entire May-October period are presented in Spang et al and not shown monthly as in Höpfner et al. (2009).

**P9L2:** 'The morphology of NAT particles is still an open question, as is the compactness of the particles (Molleker et al., 2014; Woiwode et al., 2014).' One could also mention here a recent publication by Woiwode et al., 2016.

We will add a reference to this paper which was published in ACP after we submitted our manuscript.

**P15L13:** 'However, further averaging of CALIOP backscatter (as discussed in Section 3) on 27 May (not shown) does indicate a considerably larger area of MIX1 class, and so it appears that we are dealing with the limit of the L2PSCMask detection range.' Where does this extended area lie? Below (indicating sedimentation) or in the TTE region? Maybe a figure could be added as supplemental material

The extended area of MIX1 lies mainly to the left of the STS cloud. We agree with the suggestion of adding a new figure as the effect is even more profound than the Figure 3 illustration.

**P15L20 and Fig. 9:** 'can lead to a complete removal of the available ambient HNO3 from the gas phase.'

Could you comment on the large amount of negative HNO3 values. Are those compatible with the MLS precision or does it hint to some systematic error?

We considered the underlying measurement uncertainties from MLS and the GEOS temperatures in Lambert et al 2012 . . .

> *4.3.2 Measurement uncertainties*
> *Uncertainties in the temperature analyses and nitric acid measurements contribute to the scatter of the data in Fig. 5. We note that in situ measurements of the aerosol volume density variation with temperature have shown excellent agreement with the calculated STS equilibrium curve (e.g., Drdla et al., 2003) and hence may indicate for the results here a low bias of about 1K in the GEOS-5 temperatures. The mean*

*value of the HNO3 data for the ICE classification is 0.3 ppbv, which could imply an actual total uptake of available HNO3 in the presence of ice PSCs since we are unable to rule out such a small positive bias in the MLS HNO3 measurements. The standard deviation is 0.5 ppbv, which compares well with the estimated MLS single measurement precision of 0.7 ppbv.*

We will improve the discussion of the measurement uncertainties and add a reference to these previous findings.

**P17, chapter 6:** I would like to see some discussion on the centeredness of the polar vortex around the S-pole between the different years and whether this correlates somehow with the presented data  especially related also to the central part, which is missed by the observations.

We will investigate the extent to which the polar vortex wanders around the South Pole and indicate how this interacts with the data gap poleward of 8 degrees in the revised manuscript.

**P40, Fig. 10:** Could you show here also T-Tice (e.g. in the appendix)? Does this look significantly different from TTE?

For the 32 hPa pressure level, TTE and T-TICE are shown together in Figure 13 for all years investigated including 2009. Fig 8b shows detailed cross-sections during May 2009 with clear differences between the structure of the local temperatures (T-Tice) (color shading, with T=TNAT and T=Tice + 2 K contours) and TTE.

**Technical corrections:**

We will make the corrections noted in RC1 in the revised manuscript.

**RC2 General Comments:**

This is a complicated and comprehensive over view of what may be learned by combining co-located remote measurements of gas phase HNO3 from the Aura microwave limb sounder (MLS) and the CALIOP lidar measurements of particle backscatter and depolarization. The authors combine these measurements with back trajectory calculations to address questions related to the nucleation of nitric acid trihydrate (NAT) within polar stratospheric clouds. The primary conclusion that there is significant NAT formation at temperatures well above Tice for NAT with low number concentrations and large particle sizes implies that particle observations confirming this conclusion are not possible due to the sensitivities of the lidar measurements. Thus a significant fraction of the paper is devoted to microphysical calculations of the sensitivities of the various instruments to particles of varying sizes, number concentrations, and shapes. This is necessary to ultimately rely solely on the MLS measurements of HNO3 gas phase loss as evidence of PSC formation when concomitant particle observations provide no measurable PSCs. Thus the primary conclusion on NAT nucleation requires only MLS gas phase hno3 loss and an absence of PSC detection by the CALIOP lidar. The authors present a lot of information in fairly compact forms and it is difficult to suggest any effective ways to shorten or compress the figures. The color coding can be at times difficult to fully distinguish the various categories of particle calculations and I wonder if all the aspect ratios presented are necessary. I should think the aspect ratios bounding the range are sufficient. It is not clear why the sensitivities to IR measurements at 12 um are presented. No IR measurements are used in the analysis and none are currently available for PSC measurements to my knowledge. Is this just done for completeness? In any case the slant indicating the IR measurements are somehow more sensitive to PSCs based on the graphs shown seems a bit arbitrary. The IR emission measurements will be more sensitive to larger particles since they are sensitive to volume whereas the visible scattering measurements are sensitive to aerosol cross section so more sensitive to smaller particles.

We will address the concerns of the referee in the modified manuscript. The different aspect ratios are provided to indicate that they do not change monotonically. The discussion of the IR measurements will be expanded and better justified, as also suggested by the first reviewer.

The paper should be published with minor corrections. I provide some specific comments for consideration below.

**5.6-13:** This is too much detail that will make sense to the very few who know what PSC compositions of 2 and 3 are. I would recommend a short description of what the distinction between mix1 and mix2 is and why it is important, then just explain that the separation failed to include in mix1 those observations with a scattering ratio less than 1.25. Later 1.25 is used as a threshold to identify PSC presence so overall this is a bit confusing.

We will incorporate the suggested simplified explanation into the text. However, the MIX1/MIX2 mis-classification still remains as an undocumented error in the v1 L2PSCMask operational dataset.
`http://www-calipso.larc.nasa.gov/resources/`
`calipso_users_guide/data_summaries/psc/index.php.`
Accordingly, we feel that it is necessary to document in detail the precise algorithm we actually used to correct the supplied PSC classification.

We will reorganize this section to make a better distinction between our specific usage of the v3 Level-1b standard product and the v1 Level-2 L2PSCMask.

**6.2:** 'The HNO3 and H2O values (for estimating TNAT and TICE) are assumed to be the same at all back-trajectory times.' This does not make sense. The same as what? Are they assumed to be constant from the start point of the back trajectory, so not updated along the trajectory, or ???

Yes, HNO3 and H2O are assumed to be constant from the start point of the back trajectory and we will clarify this in the revised manuscript.

**8.8:** Do the authors believe there is really uncondensed h2so4 available for uptake? If so what are they using for the ambient h2so4 gas phase mixing ratio?

An estimate of the total H2SO4 mixing ratio is required for the supercooled ternary solution thermodynamical calculations in the Carslaw et al formulation. It is set at 0.1 ppbv, which is an appropriate level for the lower stratosphere in volcanically unperturbed conditions. There is practically no H2SO4 vapor at the very low temperatures associated with PSC formation. However, binary aerosols (SBS) do evaporate at higher temperatures (see Fig 4, Carslaw et al 1997) and H2SO4 vapor is released.

**8.15:** Rt=1.25 as a threshold? This is the same value used earlier in the revised classification scheme and now here is used as a threshold to identify PSCs. Seems a bit confusing. In addition Fig. 4 c) uses 1.25 as the threshold for STS detection, so clearly NAT PSCs could appear below this level. Please clarify.

Again, we will reorganize this section to make a better distinction between our specific usage of the v3 Level-1b standard product and the v1 Level-2 L2PSCMask. Also we will indicate the sampling volumes of the respective instruments.

**8.29-31 and Fig. 4c:** This statement and the figure don't show that the IR is more sensitive than the visible to STS content. It just shows they are different. The ordinates in Fig 4c are somewhat arbitrarily chosen and the assumed thresholds are set on somewhat subjective criteria that are not fully justified and are different. Again for this paper what is the point of showing the IR results?

Only three spaceborne lidar missions have ever been flown (GLAS, CALIOP, and CATS) compared to many IR sounders over the past few decades. The purpose of including results from IR sounders is to provide context for the lidar measurements. We show that large NAT particles at low number densities are expected to be detected by IR limb sounders. There are conditions under which large size NAT PSCs are not detected with the CALIOP L2PSCMask, but can be inferred by the uptake of HNO3. Therefore it appears logical to provide some simulations to understand why this is the case and to compare against IR sounder capabilities. However, we agree that for STS, given the assumed thresholds that are based on practical empirical considerations, the IR measurement is only marginally more sensitive than the visible lidar, because a smaller HNO3 content within the STS is detectable. Now, one may argue that if the lidar signal had been averaged over a larger volume, then it would have been identified as the more sensitive

technique. However, the lidar is limited in the across track direction to around 90 m and nothing can be done about that. We have however investigated coarser averaging of the lidar data in the along-track and vertical directions in this paper. In the case of MLS, the across-track averaging (i.e. the 240 GHz antenna beam width at the tangent point) is around 8 km, the vertical field of view for HNO3 is a few km and the along-track sampling is over several hundred km. The sampling volumes of IR instruments and MLS are therefore three or more orders of magnitude larger than the typical CALIOP sampling volumes. We will discuss these sampling issues more thoroughtly in the revised manuscript.

**9.18-21:** Is this realistic? Would the large NAT particles not continue to take up hno3, given enough time, until it was all consumed, so additional uptake of hno3 is not dependent on STS?

This is a crude, but not unrealistic, snapshot of a possible STS/NAT mixture at a particular time. The Wegener-Bergeron-Findeisen process will indeed cause sequestration of HNO3 by NAT in a mixed phase STS/NAT cloud at the expense of the HNO3 in the liquid STS (*e.g. Voigt et al (2005), Nitric Acid Trihydrate (NAT) formation at low NAT supersaturation in Polar Stratospheric Clouds (PSCs), Atmos. Chem. Phys., 5, 1371-1380, doi:10.5194/acp-5-1371-2005, 2005.*). However, if the STS forms quickly by rapid cooling then the uptake of ambient HNO3 can be predominantly into STS rather than into NAT. Growth of NAT is then retarded at these low temperatures of a few K above the frost point, see Fig 4 of Voigt et al (2005). We will describe these processes more thoroughly in the revised manuscript.

**9.30-31:** Same question as above. Why does the uptake of hno3 depend on the number density of particles? Is this some sort of kinetic limit or resident time limit, i.e. particles growing too large to stay in the air space? I am not aware there is a theoretical limit to the size a NAT particle can attain given existence at T < Tnat and hno3 and h2o available.

The growth of NAT is kinetically limited, see Fig 4 of Voigt et al (2005). A high number density of NAT nuclei would ultimately lead to a NAT distribution with smaller particle sizes than a low number density since the available HNO3 is spread over a large number of particles. (*Jensen et al (2002), Impact of polar stratospheric cloud particle composition, number density, and lifetime on denitrification, J. Geophys. Res., 107(D20), 8284, doi:10.1029/2001JD000440, 2002.*). Once nucleated, a NAT particle will continue to grow, providing there is sufficient HNO3 and H2O available and T < TNAT, such that the HNO3 vapor pressure over NAT is supersaturated, until it attains its equilibrium size (reaching a radius of tens of microns). Gravitational sedimentation may cause the NAT particles to descend into a region of lower HNO3 and/or rising temperature causing evaporation rather than growth.

We will describe these processes more thoroughly in the revised manuscript.

**10.9-10:** I suppose this should be no surprise since IR emission will be sensitive to volume, R3 whereas backscatter is sensitive to cross section, R2. Thus large particles cause larger changes in volume.

This is correct. Scattering is dependent on the size parameter (x = 2 * pi* radius/wavelength).

**10.17:** How is effective radius, Reff, defined? Is this 3V/S so the ratio of the third and second

moments?

Yes.

**14.25:** I do not see a blue line, indicating Tice+2K, on any of the panels for case 3.

The area enclosed by this contour is hard to see. We will indicate specifically the location in the text.

**15.28:** I assume the authors mean 340 K (mid-June).

Yes, this should be 340 K.

**16.10:** Where is the gray shading? I do not see any. It only appears in later figures so save the explanation for then.

The gray shading appears in the daylight part of the CALIOP measurements in rows 8e and 8f. We will add to the Fig 8 caption "Gray shading indicates no observations".

**16.14:** It takes quite a few days for significant TTE to be observed at 21 hPa, nearly half the period shown, so the statement here comparing 21 and 46 hPa for the appearance of significant TTE is confusing.

We will reword this statement on the first appearence of TTE at pressures above and below 32 hPa.

**Fig 13a:** Where are the diamonds in the TTE plot?

These will be rendered in white in the revised plot to improve visibility.

---

## Author Response (AR1)

**acpd-2016-433 :: Reply to Referee Comments and Manuscript Revisions**

**Early winter Antarctic PSCs and HNO3 observed by CALIOP and MLS**

**A. Lambert, M. L. Santee and N. J. Livesey**

We thank the reviewers for their careful reading of the manuscript and appreciate their suggestions for improvements. We have addressed their comments in the following text and also revised the manuscript accordingly.

Referee Comments RC1/RC2 are in blue type.

Author Responses are in red type.

Edwards et al (1995), Selection of sounding channels for the High Resolution Dynamics Limb Sounder, Appl. Opt. 34, 7006-7018, 1995.

Hoppfner (2004), Study on the impact of polar stratospheric clouds on high resolution midIR limb emission spectra, J. Quant. Spectrosc. Radiat. Transfer, 83, 93107, 2004.

Höpfner et al (2006), Spectroscopic evidence for NAT, STS, and ice in MIPAS infrared limb emission measurements of polar stratospheric clouds, Atmos. Chem. Phys., 6, 1201-1219, doi:10.5194/acp-6-1201-2006, 2006.

Höpfner et al (2009), Comparison between CALIPSO and MIPAS observations of polar stratospheric clouds, J. Geophys. Res., 114, D00H05, doi:10.1029/2009JD012114, 2009.

Jensen et al (2002), Impact of polar stratospheric cloud particle composition, number density, and lifetime on denitrification, J. Geophys. Res., 107(D20), 8284, doi:10.1029/2001JD000440, 2002.

(L12) Lambert et al (2012), A-train CALIOP and MLS observations of early winter Antarctic polar stratospheric clouds and nitric acid in 2008, Atmos. Chem. Phys., 12, 2899-2931, doi:10.5194/acp-12-2899-2012, 2012.

Lawrence et al (2015), Comparisons of polar processing diagnostics from 34 years of the ERA-Interim and MERRA reanalyses, Atmos. Chem. Phys., 15, 3873-3892, doi:10.5194/acp-15-3873-2015, 2015.

Sembhi et al, (2012), MIPAS detection of cloud and aerosol particle occurrence in the UTLS with comparison to HIRDLS and CALIOP, Atmos. Meas. Tech., 5, 2537-2553, doi:10.5194/amt-5-2537-2012, 2012.

Spang and Remedios, (2003), Observations of a distinctive infrared spectral feature in the atmospheric spectra of polar stratospheric clouds measured by the CRISTA instrument, Geophys. Res. Lett., 30(16), 1875, doi:10.1029/2003GL017231, 2003.

Spang et al (2012), Fast cloud parameter retrievals of MIPAS/Envisat, Atmos. Chem. Phys., 12, 7135-7164, doi:10.5194/acp-12-7135-2012.

Spang et al (2016), A multi-wavelength classification method for polar stratospheric cloud types using infrared limb spectra, Atmos. Meas. Tech., 9, 3619-3639, doi:10.5194/amt-9-3619-2016, 2016.

Taylor et al. (1993), Remote sensing of atmospheric structure and composition by pressure modulator radiometry from space: The ISAMS experiment on UARS, J. Geophys. Res., 98(D6), 1079910814, 1993.

*Voigt et al* (2005), *Nitric Acid Trihydrate* (*NAT*) *formation at low NAT supersaturation in Polar* Stratospheric Clouds (PSCs), Atmos. Chem. Phys., 5, 1371-1380, doi:10.5194/acp-5-1371-2005, 2005.

Woiwode et al (2016), Spectroscopic evidence of large aspherical-NAT particles involved in denitrification in the December 2011 Arctic stratosphere, Atmos. Chem. Phys., 16, 95059532, doi:10.5194/acp-16-9505-2016, 2016.

**RC1 General Comments:**

In this paper, Lambert et al. investigate in detail the start of the PSC season in the Antarctic stratosphere with the aim to explain the early formation of nitric acid trihydrate (NAT). For this, a decade of quasi-simultaneous observations of HNO3 (and H2O) by the MLS instrument on Aura and PSC measurements by the CALIPSO lidar are utilised. The authors argue that observed early loss of HNO3 from the gas-phase without coincident detection of PSCs by CALIPSO is due to the formation of large NAT particles of low number density. These findings are corroborated by trajectory calculations indicating temperatures some Kelvin below the NAT equilibrium temperature as a prerequisite for the occurrence of such situations. This supports early NAT formation mechanisms without the necessity for the existence of ice particles, thus backing previous observations in the Arctic wintertime stratosphere. Further, a comprehensive overview of ten years of MLS and CALIPSO early-Antarctic wintertime observations shows the relative strength of NAT versus STS formation in each year. The paper is well written and conclusive. Especially I would like to point out the effort the authors have taken in the development of new methods of data presentation to support their argumentation. Thus, I strongly support its publication in ACP.

The only major points I would like the authors to consider are: (1) is the discussion of the IR limb-instruments sensitivity of any help for the flow of the arguments or does it perhaps distract the reader from the important points? (2) I miss a more in-depth discussion on measurement restrictions, like the accuracy of MLS and the lack of observations in the central part of the vortex, on the conclusions drawn.

We have addressed the concerns of the referee in the modified manuscript and improved the discussion of the sensitivity of IR limb instruments and the inherent measurement uncertainties in our analyses.

**RC1 Specific Comments:**

P4L30: 'scaled potential vorticity' Please add a reference for sPV or a short explanation.

A reference for scaled PV was added to the text.

**P5L20:** 'with single-profile precisions of 4-15% Lambert et al., 2007) and 0.7 ppbv for HNO3 (Santee et al., 2007)'. Could you also give estimates on the accuracy of H2O and HNO3 from MLS in the wintertime high-latitude regions (systematic error estimation and/or comparison with independent observations)? How strong would errors in those gases affect your following argumentation?

The text was corrected to state the MLS v4 precisions and accuracies for H2O and HNO3. We used the resulting estimated RMS errors (precision and accuracy in quadrature) to obtain the equivalent uncertainties in the derivation of TICE and TNAT by error propagation. We find that the contribution of MLS H2O and HNO3 to the RMS errors in TICE is  $\leq 0.5$  K and in TNAT is  $\leq 0.7$  K for 70–20 hPa. These values are now stated in the text in section 3.1 "Modeled uptake of HNO3, lidar backscatter and infrared extinction in PSCs" ... Errors in the calculations of these reference temperatures arising from uncertainties in the MLS H2O and HNO3 data are estimated to be  $\leq 0.5$  K for  $T_{\rm ICE}$  and  $\leq 0.7$  K for  $T_{\rm NAT}$  in the pressure range 70–20 hPa.

**P6L3:** 'Typical lower stratospheric polar values for TICE and TNAT are 188 K and 195 K, respectively.'

These values depend on pressure, H2O and HNO3. Could you indicate the ranges?

We have modified the text in section 2.3 Temperature history and relation to NAT nucleation and growth processes . . . For typical lower stratospheric polar conditions (5 ppmv H2O, 10 ppbv HNO3, and 46 hPa) values for  $T_{ICE}$  and  $T_{NAT}$  are 188 K and 195 K, respectively. Both  $T_{ICE}$  and  $T_{NAT}$  are lowered (raised) by about 2 K at 32 hPa (68 hPa). Under denitrified conditions (5 ppbv HNO3),  $T_{NAT}$  is lowered by about 1 K, and under dehydrated conditions (3 ppmv H2O),  $T_{ICE}$  is lowered by about 3 K. In denitrified and dehydrated conditions,  $T_{NAT}$  is also lowered by about 3 K.

**P8L16:** 'For a mid-infrared limb sounder operating in the window-region near 12 um we use an extinction threshold, kext =  $4 \times 10^{-5}$  km-1 based on measurements by the Improved Stratosphere and Mesosphere Sounder (Lambert et al., 1996).' Also e.g. P8L22,P10L29.

Generally, I do not understand why IR limb sounders are included in the discussion while such measurements are not used at all in the scientific argumentation of the paper. Furthermore, such a threshold depends very strongly on instrumental details like e.g. spectral noise, field-of-view and radiometric calibration accuracy. I would not assume it justified to use here thresholds determined for an instrument (ISAMS) which has not been operated in parallel to CALIPSO and MLS but to skip MIPAS and HIRDLS.

In addition, after the paper by Lambert et al, 1996, there have been findings regarding simulation of limb radiances and the importance of scattered mid-IR radiation (e.g. Höpfner, 2004) which do also impact the argumentation on sensitivity (by increasing the sensitivity for particles larger than 1 um). In conclusion, I would suggest to either go into more detail regarding the IR limb-sounding thresholds (instrumental parameters, refractive indices used for STS and NAT, forward model including scattering: : :), or, perhaps better, skip this discussion and concentrate on LIDAR and MLS.

We added further supporting material on the capabilities of IR limb sounders for aerosol/PSC detection. This is an important point, since we expect that such instruments should be able to detect the sub-visible PSCs missed by CALIOP. In the 1990's the UARS instruments ISAMS/CLAES were of course operated in a period of enhanced stratospheric aerosol loading from Mt Pinatubo. This complicates the detection of PSCs against an enhanced background aerosol loading. To provide a fair comparison for the low stratospheric background loading during the past decade we determined the extinction threshold for the ISAMS instrument for the period in 1991 before meriodional transport of volcanic sulfate had permeated the winter NH vortex. The CLAES instrument had not commenced operations during this early period. We fitted a log-normal distribution to the 12 micron ISAMS extinction measurements for latitudes poleward of 70N. The threshold is set at least three standard deviations above the mean background extinction. We find extinction thresholds of kext =  $5 \times 10^{-5}$  km-1 at 46 hPa and  $4 \times 10^{-5}$  km-1 at 32 hPa. We found that previously we had inadvertently used the wrong extinction threshold (corresponding to 32 hPa instead of 46 hPa). Hence, we have now modified the text and Figures 4, 5 and 6 to use the correct higher IR extinction threshold for 46 hPa. Note that only the lines drawn for the IR extinction have changed in these figures. Our discussion of the comparison of detection

limits has been modified accordingly. However, our prior conclusions concerning sub-visible low NAT density large particle PSCs are not materially affected.

As suggested by the referee, we also brought into the discussion the contemporary measurements from the HIRDLS instrument, which we found to have similar IR extinction thresholds during its period of operation 2005-2007. The vertical resolution of HIRDLS is about 1 km compared to 2.4 km for ISAMS.

Concerning the MIPAS instrument, since no retrieved extinction profile data are currently available, we are unable to determine the characteristics of the background extinction distribution using the method we employed for ISAMS and HIRDLS. Detection and classification of PSCs by MIPAS is carried out using ratios of spectral indices defined for carefully selected microwindows e.g. *Spang et al (2012)*. The basic PSC detection by MIPAS relies on a composition independent cloud-index parameter (CI), which is inversely related to cloud optical thickness, and defined as the ratio of the mean spectral intensities in the wavenumber ranges 788.2–796.25 cm-1 and 832.3–834.4 cm-1. The threshold used for PSC detection in L12 is CI < 4.5 (*Spang and Remedios, 2003*), and calculations by *Höpfner et al (2006)* indicate this corresponds to a volume density detection limit of 0.2–0.4  $\mu$ m3 cm-3. We now refer to this limit in the text.

A study by *Sembhi et al, (2012)*, indicates a very widely ranging spectral index threshold for the polar regions based on their annual climatological analysis. In fact, Sembhi et al conclude that ... "*The polar stratospheric thresholds show a larger degree of uncertainty over the changing seasons, meaning the polar regions would be better treated with a different approach in which time and atmospheric composition dependency are considered*". We would however expect MIPAS to achieve a comparably low threshold for PSC detection by radiance integration over a wider pass band in the 12 micron spectral window, followed by an extinction retrieval with proper accounting for the temperature and trace gas species. For example the ISAMS *Taylor et al (1993)* and HIRDLS *Edwards et al (1995)* 12 micron pass bands (50% transmission) were 817–833 (16 cm-1) and 821–836 ( 15 cm-1), respectively. These pass bands are up to a factor of 8 wider than the MIPAS 12 micron microwindow (826.4–832.3) (2.1 cm-1) and would reduce the signal/noise for aerosol detection.

The study by Hopfner (2004) showed that large particle, small number density NAT can be detected by MIPAS and we now cite this paper in the text. The specific case studied was for a particle distribution with 7 micron mode radius and number density  $2.9 \times 10^{-4}$  cm-3, which we estimate to be equivalent to a 12 micron extinction of  $1.9 \times 10^{-4}$  km-1 (practically independent of the particle aspect ratio in the infrared region).

Regarding the scattering studies carried out by Hopfner (2004), we note that since the onset of sub-visible clouds occurs during polar night the potential use of particle size determination via the scattering of solar light is not applicable here. However, scattering of tropospheric radiation into the line of sight modifies the detected line shapes of CO2 and H2O and could possibly be used for particle sizing if the complexities of the underlying surface radiation and clouds could be modeled adequately.

The characteristic NAT spectral signature near  $820 \text{ cm}^{-1}$  is washed out for particles larger than 2-3 microns ... see *Höpfner et al* (2006). Therefore, as we explained in L12 for the 2008 early winter period, the NAT spectral feature cannot help determine a positive identification for the large particle NAT.

Further to the consideration of the existence of a population of sub-visible PSCs, in a comparison of CALIOP and MIPAS PSC measurements, *Hopfner et al* (2009) found a discrepancy in the number of Antarctic PSC detections during May. In L12, we commented on the findings by Höpfner et al as follows ...

Comparisons between CALIOP and MIPAS matching pair observations by Höpfner et al. (2009) in 2006-2007 found that overall both instruments detected Antarctic PSCs in common in 85% of the matches during June-August, but only 60% in May. In particular, there were considerably more MIPAS-only PSC detections (i.e without corresponding matching detections by CALIOP) in May (25%), compared to June (8%) and July-October (< 3%). Höpfner et al. (2009) attributed these more frequent detections to the MIPAS limb-viewing geometry, which is oriented towards the pole, and results in sensitivity to PSCs lying poleward of the tangent points, but beyond the latitude limit of the CALIOP nadir view. Since PSC formation largely occurs in a region close to the South Pole in May, the MIPAS viewing geometry has a much larger effect on the PSC detection frequency at this time compared to later times when PSCs occur over a more widespread region. Note that the MLS line of sight is in the forward direction along the orbit track and is not oriented towards the pole, therefore the low HNO3 measurements are not biased by the viewing geometry.

Our expanded analysis of Antarctic early-winter PSCs now extends to a decade of measurements and provides conclusive evidence that another factor, other than viewing geometry of cloud inhomogeneity, is also at play in the lower number of Antarctic PSC CALIOP detections in May compared to MIPAS – the common occurrence in some early winters of large particle NAT clouds which are not detected by CALIOP.

Since we submitted our manuscript a new paper on the MIPAS PSC classification scheme has been published by *Spang et al (2016)*. This paper presents an intercomparison of CALIOP vs MIPAS PSC detections and statistics on the frequency of type classifications that are relevant here. Particularly, we note that only the climatological averages of PSC count statistics over the entire May-October period are presented in Spang et al and are not shown monthly as in Höpfner et al. (2009). Even with the longer term averaging, Spang et al conclude MIPAS has a higher sensitivity to PSCs since, for the cases where CALIOP detects no cloud, about 60% correspond to cloudy scenes for MIPAS. Therefore, these MIPAS/CALIOP comparisons are cited in the revised text to support our conclusion that CALIOP is not sensitive to low number density, large particle NAT.

**P9L2:** 'The morphology of NAT particles is still an open question, as is the compactness of the particles (Molleker et al., 2014; Woiwode et al., 2014).' One could also mention here a recent publication by Woiwode et al., 2016.

We added a reference to this paper which was published in ACP after we submitted our initial manuscript.

**P15L13:** 'However, further averaging of CALIOP backscatter (as discussed in Section 3) on 27 May (not shown) does indicate a considerably larger area of MIX1 class, and so it appears that we are dealing with the limit of the L2PSCMask detection range.' Where does this extended area lie? Below (indicating sedimentation) or in the TTE region? Maybe a figure could be

added as supplemental material

The extended area of MIX1 lies mainly to the left of the STS cloud, in the TTE region, but also extending below by 2–4 km We have added a new figure and revised the text as the effect is even more profound than the Figure 3 illustration.

**P15L20 and Fig. 9:** 'can lead to a complete removal of the available ambient HNO3 from the gas phase.'

Could you comment on the large amount of negative HNO3 values. Are those compatible with the MLS precision or does it hint to some systematic error?

We considered the underlying measurement uncertainties from MLS and the GEOS temperatures in Lambert et al (2012) ...

**4.3.2 Measurement uncertainties**

Uncertainties in the temperature analyses and nitric acid measurements contribute to the scatter of the data in Fig. 5. We note that in situ measurements of the aerosol volume density variation with temperature have shown excellent agreement with the calculated STS equilibrium curve (e.g., Drdla et al., 2003) and hence may indicate for the results here a low bias of about 1K in the GEOS-5 temperatures. The mean value of the HNO3 data for the ICE classification is 0.3 ppbv, which could imply an actual total uptake of available HNO3 in the presence of ice PSCs since we are unable to rule out such a small positive bias in the MLS HNO3 measurements. The standard deviation is 0.5 ppbv, which compares well with the estimated MLS single measurement precision of 0.7 ppbv.

We added the following statement ... "As shown in L12, the spread of HNO3 mixing ratios in totally denitrified regions is compatible with the MLS precision."

**P17, chapter 6:** I would like to see some discussion on the centeredness of the polar vortex around the S-pole between the different years and whether this correlates somehow with the presented data especially related also to the central part, which is missed by the observations.

The extent to which the polar vortex distorts and wanders around the South Pole can be assessed by a dynamical diagnostic such as the vortex-temperature concentricity (VTC). This hybrid temperature-vortex diagnostic (*Lawrence et al (2015), see their Eqn 1 and Figure 16*) indicates that for the southern hemisphere, the cold pool temperatures and the polar vortex are highly concentric. We have also investigated the daily scatter of temperature vs vorticity in the ERA-Interim reanalysis data, and find that the annular region 60 S to 82 S (visible to Aura and CALIOP) describes adquately the state of the region poleward of 82 S (not sampled by Aura or CALIOP).

**P40, Fig. 10:** Could you show here also T-Tice (e.g. in the appendix)? Does this look significantly different from TTE?

For the 32 hPa pressure level, TTE and T-TICE are shown together in Figure 13 for all years investigated including 2009. Fig 8b shows detailed cross-sections during May 2009 with clear differences between the structure of the local temperatures (T-Tice) (color shading, with T=TNAT and T=Tice + 2 K contours) and TTE.

**Technical corrections:**

P3L10: 'predictibility'  $\rightarrow$  'predictability' P4L8: 'superceded'  $\rightarrow$  'superseded' P5L17: 'aproximately'  $\rightarrow$  'approximately' P11L16: 'correponding'  $\rightarrow$  'corresponding' P34, caption Fig. 6: 'amibient'  $\rightarrow$  'ambient' P39, x-axis labels: Please indicate if the major tick mark labels are the beginning or the middle of the month P40, Fig. 10: Please add information on the real date (in addition to 2009d132, : : :)- at least in the caption for the first and last day

We have made the corrections noted above in the revised manuscript.

**RC2 General Comments:**

This is a complicated and comprehensive over view of what may be learned by combining co-located remote measurements of gas phase HNO3 from the Aura microwave limb sounder (MLS) and the CALIOP lidar measurements of particle backscatter and depolarization. The authors combine these measurements with back trajectory calculations to address questions related to the nucleation of nitric acid trihydrate (NAT) within polar stratospheric clouds. The primary conclusion that there is significant NAT formation at temperatures well above Tice for NAT with low number concentrations and large particle sizes implies that particle observations confirming this conclusion are not possible due to the sensitivities of the lidar measurements. Thus a significant fraction of the paper is devoted to microphysical calculations of the sensitivities of the various instruments to particles of varying sizes, number concentrations, and shapes. This is necessary to ultimately rely solely on the MLS measurements of HNO3 gas phase loss as evidence of PSC formation when concomitant particle observations provide no measurable PSCs. Thus the primary conclusion on NAT nucleation requires only MLS gas phase hno3 loss and an absence of PSC detection by the CALIOP lidar. The authors present a lot of information in fairly compact forms and it is difficult to suggest any effective ways to shorten or compress the figures. The color coding can be at times difficult to fully distinguish the various categories of particle calculations and I wonder if all the aspect ratios presented are necessary. I should think the aspect ratios bounding the range are sufficient. It is not clear why the sensitivities to IR measurements at 12 um are presented. No IR measurements are used in the analysis and none are currently available for PSC measurements to my knowledge. Is this just done for completeness? In any case the slant indicating the IR measurements are somehow more sensitive to PSCs based on the graphs shown seems a bit arbitrary. The IR emission measurements will be more sensitive to larger particles since they are sensitive to volume whereas the visible scattering measurements are sensitive to aerosol cross section so more sensitive to smaller particles.

We have addressed the concerns of the referee in the modified manuscript. The different aspect ratios are provided to indicate that they do not change monotonically. The discussion of the IR measurements have been expanded and better justified, as also suggested by the first reviewer.

The paper should be published with minor corrections. I provide some specific comments for consideration below.

**5.6-13:** This is too much detail that will make sense to the very few who know what PSC compositions of 2 and 3 are. I would recommend a short description of what the distinction between mix1 and mix2 is and why it is important, then just explain that the separation failed to include in mix1 those observations with a scattering ratio less than 1.25. Later 1.25 is used as a threshold to identify PSC presence so overall this is a bit confusing.

We have incorporated the suggested simplified explanation into the text. However, the MIX1/MIX2 mis-classification still remains as an undocumented error in the v1 L2PSCMask operational dataset.

http://www-calipso.larc.nasa.gov/resources/ calipso\_users\_guide/data\_summaries/psc/index.php. Accordingly, we feel that it is necessary to document in detail the precise algorithm we actually used to correct the supplied PSC classification. We made a better distinction between our specific usage of the gridded v3 Level-1b standard product and the v1 Level-2 L2PSCMask.

**6.2:** 'The HNO3 and H2O values (for estimating TNAT and TICE) are assumed to be the same at all back-trajectory times.' This does not make sense. The same as what? Are they assumed to be constant from the start point of the back trajectory, so not updated along the trajectory, or ???

Yes, HNO3 and H2O are assumed to be constant from the start point of the back trajectories and we have clarified this in the revised manuscript.

**8.8:** Do the authors believe there is really uncondensed h2so4 available for uptake? If so what are they using for the ambient h2so4 gas phase mixing ratio?

An estimate of the total H2SO4 mixing ratio is required for the supercooled ternary solution thermodynamical calculations in the Carslaw et al formulation. It is set at 0.1 ppbv, which is an appropriate level for the lower stratosphere in volcanically unperturbed conditions. There is practically no H2SO4 vapor at the very low temperatures associated with PSC formation. However, binary aerosols (SBS) do evaporate at higher temperatures (see Fig 4, Carslaw et al 1997) and H2SO4 vapor is released. We have modified the text accordingly.

**8.15:** Rt=1.25 as a threshold? This is the same value used earlier in the revised classification scheme and now here is used as a threshold to identify PSCs. Seems a bit confusing. In addition Fig. 4 c) uses 1.25 as the threshold for STS detection, so clearly NAT PSCs could appear below this level. Please clarify.

We have made a better distinction between our specific usage of the gridded v3 Level-1b standard product and the v1 Level-2 L2PSCMask. Also we now indicate the sampling volumes of the respective instruments. Following Pitts et al (2011), the  $R_T = 1.25$  threshold is used only to detect STS, whereas STS/NAT mixture detection uses only the perpendicular backscatter threshold. In this way, whenever there is a detectable presence of solid NAT (non-spherical, depolarizing) mixed with ambient STS (liquid, spherical, non-depolarizing) that pixel is always flagged as either MIX1 or MIX2. In the case of MIX1 the total backscatter can be below  $R_T = 1.25$ .

**8.29-31 and Fig. 4c:** This statement and the figure don't show that the IR is more sensitive than the visible to STS content. It just shows they are different. The ordinates in Fig 4c are somewhat arbitrarily chosen and the assumed thresholds are set on somewhat subjective criteria that are not fully justified and are different. Again for this paper what is the point of showing the IR results?

Only three spaceborne lidar missions have ever been flown (GLAS, CALIOP, and CATS) compared to many IR sounders over the past few decades. The purpose of including results from IR sounders is to provide context for the lidar measurements. We show that large NAT particles at low number densities are expected to be detected by IR limb sounders. There are conditions under which large size NAT PSCs are not detected with the CALIOP L2PSCMask, but can be inferred by the uptake of HNO3. Therefore it appears logical to provide some simulations to understand why this is the case and to compare against IR sounder capabilities. Since we used a practical empirical method (see response to RC1) to determine the background IR aerosol extinction thresholds by fitting to observations (i.e. without assuming instrument parameters) we feel these are indeed justified.

However, we agree that for STS, given the assumed thresholds, the IR measurement is only marginally more sensitive than the visible lidar (because a smaller HNO3 content within the STS is detectable). Now, one may argue that if the lidar signal had been averaged over a larger volume, then it would have been identified as the more sensitive technique. However, the lidar is limited in the across track direction to around 90 m and nothing can be done about that. We have however investigated coarser averaging of the lidar data in the along-track and vertical directions in this paper. In the case of MLS, the across-track averaging (i.e. the 240 GHz antenna beam width at the tangent point) is around 8 km, the vertical field of view for HNO3 is a few km and the along-track sampling is over several hundred km. The sampling volumes of IR instruments and MLS are therefore three or more orders of magnitude larger than the typical CALIOP sampling volumes.

The across-track averaging of CALIOP is about 0.09 km for CALIOP. Using these values we can construct some nominal sampling volumes that demonstrate the much larger limb sounder sampling volume.

(1) CALIOP gridded L1B data : 50 x 0.5 x 0.09 = 2.25 km3
 (2) CALIOP lowest resolution L2PSCMask: 135 x 0.18 x 0.09 = 2.19 km3
 (3) CALIOP highest resolution L2PSCMask: 5 x 0.18 x 0.09 = 0.081 km3
 (4) CALIOP regridded L2PSCMask: 165 x 2.7 x 0.09 = 40.1 km3
 (5) MLS HNO3 : 400 x 4 x 8 = 12800 km3

We discuss these sampling issues more thoroughtly in the revised manuscript.

**9.18-21:** Is this realistic? Would the large NAT particles not continue to take up hno3, given enough time, until it was all consumed, so additional uptake of hno3 is not dependent on STS?

**9.30-31:** Same question as above. Why does the uptake of hno3 depend on the number density of particles? Is this some sort of kinetic limit or resident time limit, i.e. particles growing too large to stay in the air space? I am not aware there is a theoretical limit to the size a NAT particle can attain given existence at T < Tnat and hno3 and h2o available.

We described these processes more thoroughly in the revised manuscript.

This is a crude, but not unrealistic, snapshot of a possible STS/NAT mixture at a particular time. The Wegener-Bergeron-Findeisen process will indeed cause sequestration of HNO3 by NAT in a mixed phase STS/NAT cloud at the expense of the HNO3 in the liquid STS (*e.g. Voigt et al* (2005). However, if the STS forms quickly by rapid cooling then the uptake of ambient HNO3 can be predominantly into STS rather than into NAT. Growth of NAT is then retarded at these low temperatures of a few K above the frost point, see Fig 4 of Voigt et al (2005).

The growth of NAT is kinetically limited, see Fig 4 of Voigt et al (2005). A high number density

of NAT nuclei would ultimately lead to a NAT distribution with smaller particle sizes than a low number density since the available HNO3 is spread over a large number of particles. (Jensen et al (2002)). Once nucleated, a NAT particle will continue to grow, providing there is sufficient HNO3 and H2O available and T < TNAT, such that the HNO3 vapor pressure over NAT is supersaturated, until it attains its equilibrium size (reaching a radius of tens of microns). Gravitational sedimentation may cause the NAT particles to descend into a region of lower HNO3 and/or rising temperature causing evaporation rather than growth.

**10.9-10:** I suppose this should be no surprise since IR emission will be sensitive to volume, R3 whereas backscatter is sensitive to cross section, R2. Thus large particles cause larger changes in volume.

This is correct. Scattering is dependent on the size parameter (x = 2 \* pi\* radius/wavelength). No changes were made to the text.

**10.17:** *How is effective radius, Reff, defined? Is this 3V/S so the ratio of the third and second moments?*

Yes. No changes were made to the text.

**14.25:** *I* do not see a blue line, indicating Tice+2K, on any of the panels for case 3.

The area enclosed by this contour is hard to see. We have indicated specifically the location in the text.

**15.28:** *I* assume the authors mean 340 K (mid-June).

Yes, this should be 340 K.

**16.10:** Where is the gray shading? I do not see any. It only appears in later figures so save the explanation for then.

The gray shading appears in the daylight part of the CALIOP measurements in rows 8e and 8f. We added to the Fig 8 caption "Gray shading in (e,f) indicates that the CALIOP observations are in daylight".

**16.14:** It takes quite a few days for significant TTE to be observed at 21 hPa, nearly half the period shown, so the statement here comparing 21 and 46 hPa for the appearance of significant TTE is confusing.

We have reworded this statement on the first appearence of TTE at pressures above and below 32 hPa. "Significant TTE... gradually increases in area, and expands in vertical extent rapidly to the 46 hPa level and eventually to the 68 and 21 hPa levels with a delay of about a week.

Fig 13a: Where are the diamonds in the TTE plot?

These are now rendered in white in the revised plot to improve visibility.

[revised manuscript text omitted]

---

## Author Response (AR2)

**acpd-2016-433 :: Reply to Editor Comments**

Co-Editor Decision: Publish subject to technical corrections (21 Nov 2016) by Farahnaz Khosrawi Comments to the Author:

- P1, L22: skip " hereafter L12" and write in the following Lambert et al. (2012) instead of L12.

done

- P2, L6: Same as above for Pitts et al. (2013). Using the reference itself instead of the abbreviation is much easier to read.

P8, L17: "kelvin" → "Kelvin".

non-capitalized is the correct form for unit names e.g. newton, joule, watt, etc

P11, l30: Typing error in "Höofner" should be corrected.

done

P18, 29: adquetaly → adequately.

done

P19, l13: Please give the full reference, thus add the year to the reference of Höpfner et al.

ok, however it is standard to omit the year to indicate that the previous citation is still within scope

P20, l8: One closing bracket is obsolete.

done

[revised manuscript text omitted]